# Understanding the stress field at the lateral termination of a thrust fold using generic geomechanical models and clustering methods

Anthony Adwan[1, a], Bertrand Maillot[1], Pauline Souloumiac[1], Christophe Barnes[1], Christophe Nussbaum[2], Meinert Rahn[3], and Thomas Van Stiphout[3]

[1]CY Cergy Paris Université, Laboratoire Géosciences et Environnement Cergy (GEC), 1, rue Descartes, 95000 Neuville-sur-Oise, France

[a]Now at IRSN, Institute for Radiological Protection and Nuclear Safety, 31 Av. de la Division Leclerc, 92260 Fontenay-aux-Roses, France

[2]Swisstopo, Federal Office of Topography, Route de la Gare 63, CH-2882 Saint-Ursanne, Switzerland

[3]Swiss Federal Nuclear Safety Inspectorate, Brugg, Switzerland

**Correspondence:** Anthony Adwan (anthony.adwan@irsn.fr)

**Abstract.**

This study employs numerical simulations based on the Limit Analysis (LA) method to calculate the stress distribution in a model that includes a basal detachment, featuring the lateral termination of a generic fault under compression. We conduct 2500 2D and 500 3D simulations, varying basement and fault friction angles, to analyze and classify the results into clusters representing similar failure patterns to understand the stress fields. Automatic fault detection methods are employed to identify the number and positions of fault lines in 2D and fault surfaces in 3D. Clustering approaches are utilized to group the models based on the detected failure patterns. For the 2D models, the analysis reveals three primary clusters and five transitional ones, qualitatively consistent with the critical Coulomb wedge theory and the influence of inherited structural and geometric aspects over rupture localization. In the 3D models, four different clusters portray the lateral prolongation of the inherited fault. High stress magnitudes are detected between the compressive boundary and the activated or created faults, and at the root of the inherited active fault. Tension zones appear near the outcropping surface relief while stress decreases with depth at the footwall of the created back-thrusts. A statistical, cluster-based stress field analysis indicates that for a given cluster, the stress field mainly conserves the same orientations, while the magnitude varies with changes in friction angles and compressive field intensity, except in failure zones where variations are sparse. Small parametric variations could lead to significantly different stress fields, while larger deviations might result in similar configurations. The comparison between 2D and 3D models shows the importance of lateral stresses and their influence on rupture patterns, distinguishing between 3D analysis and 2D cross-sections. Lastly, despite using small-scale models, stress field variations over a span of a couple of kilometers are quite large.

## 1 Introduction

The existing three-dimensional stress state serves as the foundation for understanding fault behaviors (Zoback, 1992; Segall and Fitzgerald, 1998; Suppe, 2007; Walsh III and Zoback, 2016; Brodsky et al., 2020), as well as for subsurface site studies (Terzaghi, 1943; Zoback, 2010). An early investigation by Lieurance (1933) focused on the rock stress state at the bottom tunnel

of the Hoover Dam and introduced the first stress measurement method based on surface relief. Since then, stress measurement techniques have undergone significant evolution. This evolution encompasses various advancements, ranging from the introduction of flat jacks (Mayer and Marchand, 1951; Tmcelin, 1951; Panek and Stock, 1964), hydraulic fracturing (Haimson and

Fairhurst, 1967), borehole breakouts (Bell and Gough, 1979), overcoring (Martin et al., 1990; Martin and Simmons, 1993), and drilling-induced tensile fracturing (Brudy et al., 1997), up to the attempts to utilize numerical modeling and simulations (Jing, 2003).

   Throughout this evolutionary trajectory, the primary concern was obtaining and analyzing the stress tensor, specifically the maximum horizontal stress ($S_H$) (Tingay et al., 2005). However, while data regarding $S_H$ orientation is readily accessible

(e.g., Heidbach et al., 2018, World Stress Map), stress magnitude values are sparse. The only available compilations are often presented in the form of misleading depth-related gradients generally derived from the minimum horizontal stress ($S_h$) values (Gunzburger and Cornet, 2007). Nevertheless, possessing extensive knowledge concerning in-situ stresses is imperative, which explains the practice of combining geomechanical models with site investigation (Jing, 2003; Reiter and Heidbach, 2014; Bergen et al., 2019; Ziegler and Heidbach, 2020).

We present a new approach for analyzing rupture occurrence, and evaluating stress fields in both 2D and 3D models. Our methodology involves a parametric sensitivity analysis based on the theory of Limit Analysis (LA) (Drucker et al., 1952; Salençon, 1974, 1983). The current study employs kilometric-scale models, in both 2D and 3D configurations. The models represent the termination of a fault-cored anticline, which extends into a wedge at the rear (Figure 1). The 3D aspect of such a triangular wedge can be found in many thrust fold regions, such as the Zagros folds (Berberian, 1995; Jahani et al., 2009), and

has previously been explored through both sandbox experiments (e.g., Graveleau et al., 2012), and numerical investigations (e.g., Conin et al., 2012; Ruh et al., 2013; Buiter et al., 2016). Yet, to the best of our knowledge, no prior research considered the existence of inherited faults nor delved into the complexity of the stress field at the scale of the lateral termination of a fault-cored anticline.

   We decide to vary both basal and inherited fault friction angles simultaneously in 2500 iterations for the 2D model and 500

iterations for the 3D model. Each of these iterations constitutes a unique simulation. Our objective is to evaluate how changes associated with the two varying parameters impact rupture propagation at the termination of an inherited fault and the stress field at the onset of rupture. To analyze this extensive dataset, we divide it into distinct, homogeneous categories and subsets using clustering techniques. We employ the automatic fault detection method proposed by Adwan et al. (2024) and define the number of faults detected, their locations and their geometry as our main comparison criteria also referred to as descriptors.

For the 2D simulations, we adopt the k-means algorithm (MacQueen, 1967) while manual clustering proved sufficient for the 3D model. Within each resulting cluster, we assess the stress field by analyzing parameters such as the pressure also referred to as the mean stress ($p$), equivalent deviatoric stress ($q$), and the orientation of the principal stresses.

   In the following sections, we begin by introducing the geologic model and preparing the setup for the study. Afterwards, we divide our exploration into two distinct parts. First, we tackle the 2D analysis, where we present the clustering results with

the obtained rupture configurations. Afterwards, we compare the results to the Critical Coulomb Wedge (CCW) theory and perform a detailed statistical examination of the associated stress fields. We then extend our investigation to 3D simulations

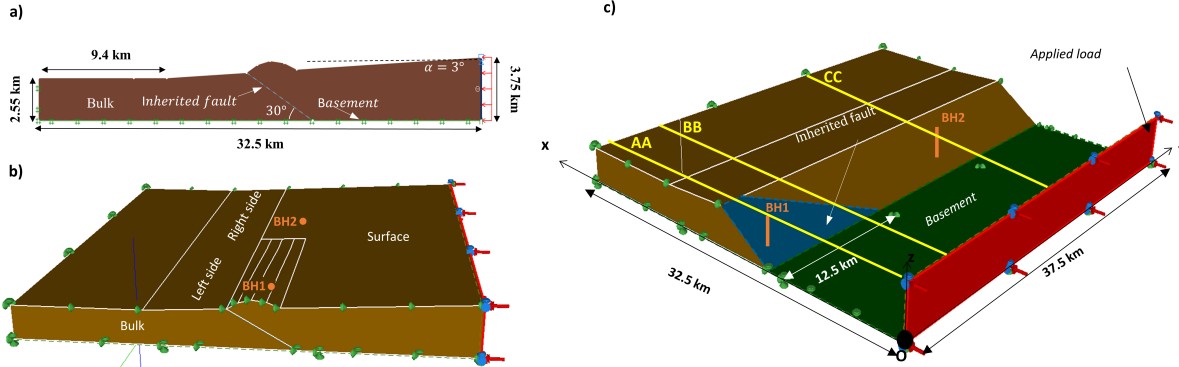

**Figure 1.** Model overview. a) 2D case (length: 32.5 km). The green color represents the basement, while the inherited fault is denoted by a dashed blue line. Red arrows indicate the unknown distributed load applied at the back wall. b and c) 3D case (length: 32.5 km, width: 37.5 km). The basement is also depicted in green, and the lateral terminating fault, spanning 12.5 km, is shown in blue. Red arrows indicate the applied external load at the back wall. The three 2D cross-sections are represented in yellow while the two boreholes are represented in orange.

following the same analysis. We juxtapose the observations from both the 2D and 3D simulations by performing a set of 2D cross-sections. We also investigate the setting around two distinct boreholes in the 3D simulations and we evaluate the resultant stress fields. Finally, we discuss our findings and give adequate conclusions on this stress analysis methodology.

## 2   Models setup and Limit Analysis implementation

The model developed in this study consists of both 2D and 3D configurations. It corresponds to the lateral termination of a partially buried fault-cored anticline (Figure 1). It is formed by an accretionary wedge at the back exhibiting a 3° topographic slope ($\alpha$). It has a length of 32.5 km, a width of 37.5 km and is formed by a uniform bulk Coulomb material with a specific weight of 25.5 kN/m² (considering a volumic mass of 2.6 kg/m³ and a gravitational acceleration of 9.81 m/s²), a cohesion ($c_{Bulk}$) of 15 MPa and an internal friction angle ($\phi_{Bulk}$) of 30° (Table 1). The basement is characterized as planar, cohesionless, with a friction angle denoted $\phi_{Basement}$. Additionally, the model incorporates a cohesionless inherited fault plane with an internal fault friction angle denoted as $\phi_{Fault}$ and a 30° dip angle. In the 3D variant, the fault terminates laterally over a distance of 12.5 km (Figure 1c).

Our primary objective is to investigate the influence of varying basal and fault friction angles on the stress field. For simplicity, we utilize stress and strain fields derived from the theory of limit analysis (LA) (Drucker et al., 1952; Salençon, 1974, 1983). LA is a widely used method in civil and geotechnical engineering to determine the maximum load a structure or model can withstand before failure. The method identifies the solution within bounds: the upper bound is associated with an optimized virtual velocity field at failure (kinematic approach), and the lower bound is associated with an optimized balanced stress field following a specified failure criterion (static approach). In LA, stress and strain are independently determined by their respec-

tive approaches, eliminating the need for a stress-strain relationship. This is why LA does not require the definition of elastic parameters. It only relies on the principle of maximum work and on the convexity of the yield criterion (here, the Coulomb criterion). This simplicity reduces the number of parameters needed compared to methods that solve the complete mechanical problem often using FEM. Despite this simplicity, LA remains a robust method capable of handling complex geometries and loading conditions.

The way the lower bound approach of Limit analysis works is by assuming a statically admissible stress field (i.e., verifying boundary conditions and equilibrium), verifying that it respects the Coulomb criterion everywhere, and computing the associated external load that it can withstand. This is a lower bound on the exact external load at failure. Then, through an optimization procedure with respect to the stress field, we compute the maximum lower bound of the external load. Thus, any external load that is lower than the maximum lower bound will be sustained without any plastic failure in the model. The upper bound approach (although not used in this paper) is approximately symmetric with the lower bound: it involves the optimization of an upper bound of the external load with respect to kinematically acceptable velocity fields. The exact external load remains unknown in limit analysis: it is bounded by the optimized lower and upper bounds. Their difference gives an estimation of the uncertainty on its value at the onset of failure.

Analytic solutions for LA problems are mainly available for simple geometries, necessitating numerical implementations for more complex cases. Initially, linear programming techniques were employed to conserve the LA bounds by approximating the Mohr-Coulomb failure criterion (Sloan, 1988, 1989). The advantages of these linear formulations were particularly evident for lower bound calculations. The use of linear shape functions ensured that normal and shear stresses were consistent on both sides of existing discontinuities, maintaining equilibrium. However, this linear scheme is limited and highly simplified, highlighting the need for nonlinear methodologies. With current numerical advancements, nonlinear implementations are achieved through improved finite-element limit analysis formulations (FELA). Each node in the limit analysis mesh is exclusive to a mesh element. This particularity allows statically admissible stress and kinematically admissible velocity discontinuities along edges of adjacent elements. An example of this formulation is found in the geotechnical software OPTUM G2-G3 (Krabbenhøft et al., 2007; Krabbenhøft and Lyamin, 2014) used in this study.

We follow the steps presented in Adwan et al. (2024) and we configure our simulations to replicate a compression regime:

- At the rear, we establish a rigid plate, hereafter referred to as the back wall, and we apply an unknown load along the x-axis perpendicular to this back wall. We restrict any rotation or vertical movement, only allowing a "push movement" following the same direction as the applied load. Between this rigid back wall and the bulk material, a contact interface with identical frictional properties as the bulk material is maintained.

- At the frontal edge and on both lateral sides (for the 3D case), normal supports are defined. These supports exclusively counteract forces perpendicular to the edge planes, preventing any movement in that direction. This also means that the movements parallel to the edges are free.

- The basement is considered fixed, meaning that all movements in all directions are denied. In order to allow slip under the applied external load, a frictional plane is implemented as a contact surface between the bulk and the defined fixed basement. This plane is cohesionless and follows the Coulomb's friction law with internal friction angles varying within [0, 25°] for the

2D model and [0, 20°] for the 3D model (Table 1).

- The inherited faults are also defined using frictional planes. They are cohesionless with internal friction angles varying within [0, 36°] for the 2D models and [0, 20°] for the 3D models (Table 1).

For the 2D configuration, 2500 simulations are achieved progressively by generating eight different batches of random friction angle values, depending on the parts deemed necessary to explore in the defined parameter space (as seen in Figure 2c) with distinct batches of simulations exploring the full friction angles space). Regarding the 3D model, simulations are conducted with a single batch of random values based on a logarithmic transformed normal distribution function centered at 10° with a standard deviation of 2. The use of such a function is to ensure that the obtained values are restricted to the defined interval with a focus on the values located at its center.

To perform hundreds of simulations, adequate meshing configurations are necessary. Following extensive convergence tests, using a computer equipped with 32 GB RAM and an 8 GB dedicated graphics card featuring an AMD Radeon chipset, a comparative assessment of the results derived from upper, lower, and mixed bound calculations revealed that the most favorable convergence occurred when the number of elements exceeded 30,000 tetrahedral elements for 3D and 10,000 triangular elements for 2D. Additionally, the time required to execute a single simulation ranged from 5 minutes to over 4 hours (for 100,000 tetrahedral elements), depending on the chosen configuration.

Finally, the chosen configuration for this study employs a lower bound limit analysis calculation using a triangular mesh of 10,000 elements for 2D, and a mixed bound analysis using 40,000 tetrahedral elements for 3D. The mixed bound analysis is a more efficient (less time-consuming) optimization procedure following the mixed principles (Zouain et al., 1993; Borges et al., 1996; Krabbenhøft et al., 2007; Adwan et al., 2024). Rather than calculating precise bounds, these advanced principles consider both stress and velocities as primary variables. By constructing finite element discretizations, the requirements of the upper and lower bound theorems are combined, offering compromise solutions that are often closer to the exact solution than the individual bounds. The entire workflow, encompassing model creation, stress tensor generation, and result processing, is automated using dedicated MATLAB codes.

**Table 1.** Simulation Parameters for the 2D and 3D LA modeling using Optum G2/G3

| Simulation Type (Count) | Bulk | Basement | Inherited Fault |
|---|---|---|---|
| 2D (2500) | $\gamma_{Bulk} = 25.5\,\mathrm{kN/m^3}$ | $c_{Basement} = 0\,\mathrm{MPa}$ | $c_{Fault} = 0\,\mathrm{MPa}$ |
| | $c_{Bulk} = 15\,\mathrm{MPa}$ | $\phi_{Basement}=[0, 25°]$ | $\phi_{Fault}=[0, 36°]$ |
| | $\phi_{Bulk} = 30°$ | | |
| 3D (500) | $\gamma_{Bulk} = 25.5\,\mathrm{kN/m^3}$ | $c_{Basement} = 0\,\mathrm{MPa}$ | $c_{Fault} = 0\,\mathrm{MPa}$ |
| | $c_{Bulk} = 15\,\mathrm{MPa}$ | $\phi_{Basement}=[0, 20°]$ | $\phi_{Fault}=[0, 20°]$ |
| | $\phi_{Bulk} = 30°$ | | |

## 3 Coulomb Critical Wedge (CCW) theory

The Coulomb wedge theory explores the stability and the deformation of an accretionary wedge sliding along a frictional basement (Davis et al., 1983). The wedge attains a critical geometry reaching a point of internal stress where the entire structure is on the verge of Coulomb failure. Following stress equilibrium, Coulomb yielding criterion and frictional properties, two critical wedge states can be identified. In the first state, the wedge is on the brink of failure in horizontal compression while the second one considers failure in horizontal extension. Since we focus our study over compressive behaviors using a cohesive bulk material, we rely on the generalized solution based on Mohr's construction (Lehner, 1986; Cubas et al., 2013). The critical taper angle is determined by the angle $\Psi_b$, between the direction of the maximum principal stress $\sigma_1$ and the base of the wedge, and the angle $\Psi_0$, between the direction of $\sigma_1$ and the top of the wedge. It can be written as follow:

$$\alpha + \beta = \Psi_b - \Psi_0, \tag{1}$$

where $\beta$ is the basement dip angle (equal to zero in our simulations). And since we do not consider pore pressure in this study, $\Psi_b$ and $\Psi_0$ are determined through:

$$\Psi_b = \frac{1}{2}\arcsin\left(\frac{\sin\phi_{Basement}}{\sin\phi_{Bulk}}\right), \tag{2}$$

$$\Psi_0 = \frac{1}{2}\arcsin\left(\frac{\sin\alpha}{\sin\phi_{Bulk}}\right). \tag{3}$$

The critical relation between both $\alpha$ and $\beta$ (1) forms a critical envelope representing three distinct states. Inside the envelope the wedge is considered stable and can slide along the basement without any internal deformation. Outside the envelope, the wedge presents internal deformation and is thus considered unstable. On the envelope itself, the wedge is on the verge of collapse and is considered in one of the two critical states previously defined.

## 4 2D analysis

### 4.1 2D rupture pattern and clustering

Following the methodology presented in Adwan et al. (2024), the evaluation of rupture using LA revolves around assessing one of three criteria: the normalized distance to the Coulomb failure criterion ($d_n$), an equivalent von Mises strain expression ($J_{vM}$), or a distance-to-strain ratio $R_{crit}$:

$$R_{crit} = \frac{d_n}{J_{vM}}. \tag{4}$$

In this study we adopt the following $d_n$ expression:

$$d_n = \frac{d\cos(\phi)}{c}, \tag{5}$$

where

$$d = \frac{|\sigma_1 + \sigma_3|}{2} sin(\phi) + c\cos(\phi) - \frac{|\sigma_1 - \sigma_3|}{2}. \tag{6}$$

$\phi$ is the internal friction angle and $c$ the cohesion (for faults, bulk, or basement). $\sigma_1$ and $\sigma_3$ are the maximum and minimum principal stresses respectively. In a more straightforward approach, and according to the Coulomb criterion, rupture occurs
wherever $d_n = 0$. However, in this case and following the LA optimization algorithm, there is strictly only one meshing element verifying this criterion with $d_n$ values over its constituting nodes equal to zero. Following this reasoning, and in order to detect incipient faults we need to consider not just the zero valued nodes but also the ones with very small $d_n$ values. For this purpose Adwan et al. (2024) introduced a Cauchy distribution scale parameter $\delta$. Through the use of this parameter, imminent failure zones can be isolated and fault lines can be extracted using image or data processing techniques. Integrating this parameter
with the previously defined $d_n$ criterion leads to the following transformed form:

$$Tr_{d_n} = \frac{1}{1 + (\frac{d_n}{\delta})^2}. \tag{7}$$

The same transformation can be applied for $J_{vM}$ and $R_{crit}$, but it is already explained in Adwan et al. (2024) and so we will not delve into this aspect any further. We employ $\delta$ values of 0.002, 0.02, and 0.1 for the three criteria ($d_n$, $J_{vM}$, $R_{crit}$) respectively. The line detection algorithm applied over the 2D dataset of 2500 simulations (see Figure 3) is based on the Radon
transform (Radon, 1917). The number of detected lines and their positions for each simulation are conserved. Results obtained from all three criteria are consistent, validating our choice of $d_n$ as our primary criterion. The detected lines are regrouped in pairs following their intersection with the basement. Each pair is characterized by two lines having a similar position with opposite dipping angle signs. For each simulation, such ramp/back-thrust system is henceforth referred to as 'V'. The position of each 'V' ($x_{v_i}$) is determined through averaging both x-coordinates obtained from the intersections between the basal level
and the detected lines (see Figure 2a). The x-axis origin is at the back wall.

The number of 'V's obtained serves as the first clustering descriptor. Our simulations are thus divided into two subsets: 2472 simulations with one 'V' (see Figure 3) and 28 simulations with 2 'V's. Subsequently, a second clustering step is applied to further distinguish our simulations. This time, we consider the retained position of each 'V' as the primary descriptor, and we explore different clustering algorithms while automatically searching for the optimal number of clusters. Both the
k-means and the agglomerative hierarchical clustering methods yield the same number of clusters, when optimized by either the Davies-Bouldin index (Davies and Bouldin, 1979) or the Silhouette coefficient (Rousseeuw, 1987). For the set with only one 'V' system, the clustering method results in three distinct clusters (C1, C2, and C3), as illustrated in Figure 2b. Regarding simulations with two 'V' systems, we obtain a set of five distinct clusters (C4, C5, C6, C7, and C8). The resulting clusters with their characterized rupture patterns are regrouped in Figure 3.

In order to interpret these clusters, we look at their distribution in function of the varying fault and basal friction angles (Figure 2c). Starting with a fault friction angle higher than 33°, the inherited fault is locked and rupture can be determined using the CCW theory. Based on the homogeneous Coulomb material parameters adopted, while considering the topographic and basal slopes, the critical basement friction angle for an $\alpha$ of 3° and a $\beta$ equal to zero is found to be 9.9° (vertical black

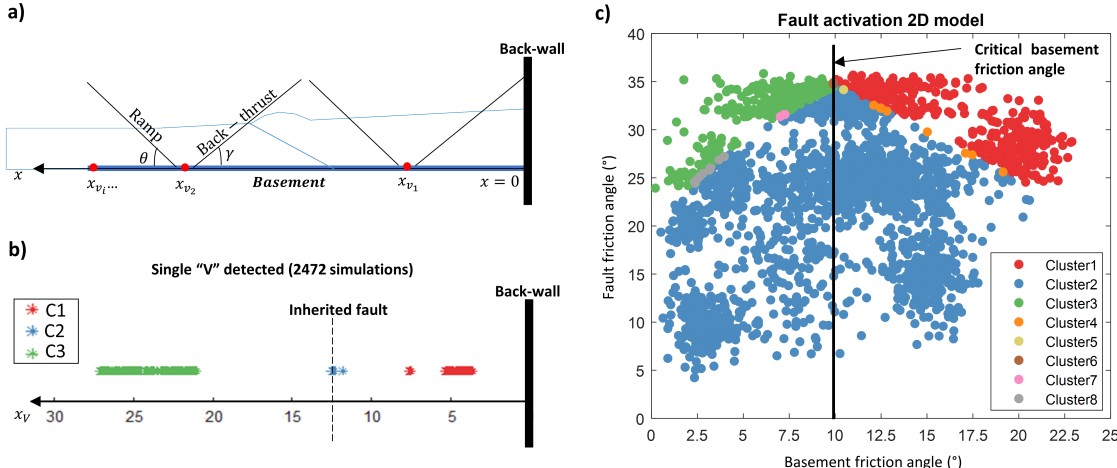

**Figure 2.** a) Defining clustering descriptors: The detected lines are illustrated in black, and the position of the ramp/back-thrust system ('V') is indicated by a red point. b) Graph displaying the distribution of obtained 'V' positions along the x-axis for the single 'V' cases. Three distinct clusters are identified and represented by different colors. Cluster C1, located at the back, is shown in red, while C2 is in blue, and C3 is in green. c) Partition graph of each simulation based on both basement and fault friction angles. Each simulation is color-coded according to its respective cluster. The position of the critical basement friction angle of 9.9° is represented by a black vertical line.

line, Figure 2c). For basal friction angles higher than this limit, the wedge is unstable and rupture is localized at the back as
spotted in C1 with 373 simulations (Figure 3). On the other hand, lower basal friction angles lead to a stable wedge where the basement is activated and rupture is at the front of the model, as evident in C3 with 247 simulations (Figure 3).

    The wedge in this study presents an outcropping relief altering the location of thrusts (Cubas et al., 2008). This effect is visible in Figure 2b where C1 is formed by two clearly distinct 'V' positions at the back. In fact, despite C1 primarily consisting of simulations showing the formation of a 'V' starting at the back wall (intersection between the back wall and
the created back-thrust), it also includes some simulations where the fault extends to the back edge of the relief (intersection between the rear side of the relief and the created ramp). This transitional pattern is identified in C5 (Figure 3). The same can be stated for C3 considering both simulations presenting a 'V' starting at the frontal edge of the relief and simulations with a 'V' located at the transition between the wedge termination and the planar frontal surface. The distinction between these two systems is not straightforward, as the model's length makes it challenging to clearly separate the 'V' positions (Figure 2b). A
more detailed analysis could have been conducted by examining the intersection of each 'V' with the model's surface to apply a third sub-clustering step, but it was deemed unnecessary in this study.

    The two simulations of basal friction angles with values very close to the critical angle, found in C6, result in an immediate transition from the back to the front, without activating the inherited fault (Figure 3). In these simulations, the model is in a critical state and we are located on the CCW envelope itself.

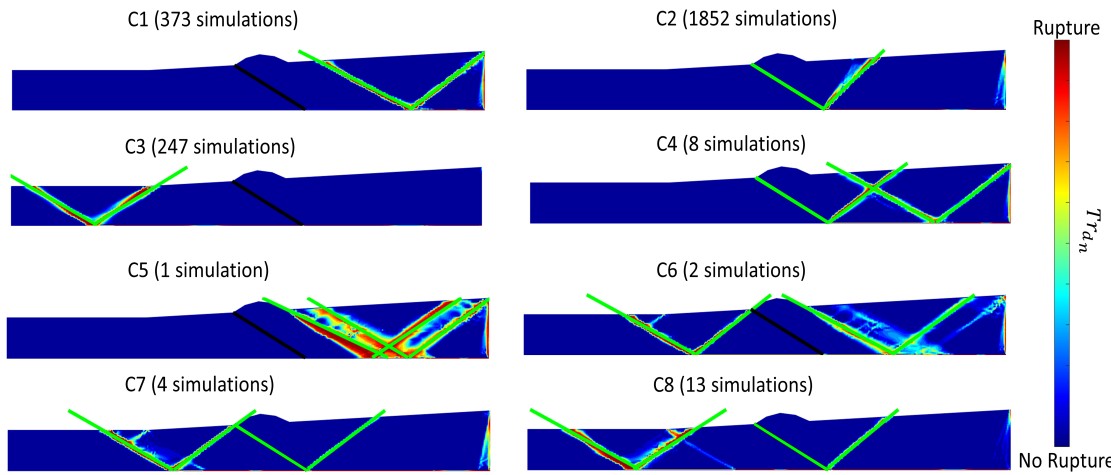

**Figure 3.** Examples of the Cauchy transformed, normalized distance to the Coulomb failure criterion ($d_n$) and the fault lines detected (in green) through the Radon Transform for all eight clusters. Values of $Tr_{d_n}$ closer to zero are shown in red, while others are displayed in blue.

As the fault friction angle decreases following the interval [24°, 33°] (Figure 2c), the influence of the inherited fault becomes more detectable. For higher values of basement friction angle, C1 is dominant. With the decrease in the basement angle values, the inherited fault starts to be activated as observed in C4, presenting 8 simulations, where such activation is detected in addition to the newly created fault system at the back wall (Figure 3). Afterwards, this fault is solely activated in C2 with 1852 simulations, until attaining basal friction angles lower than the critical limit. At this point, C7 and C8 with 4 and 13 simulations respectively, separate the transition between C2 and C3 following the above indicated geometric perturbations (Figure 3). Finally, fault friction angles lower than 24° result in a single C2 rupture pattern regardless of the basal friction angle.

## 4.2 2D stress analysis

To enhance our comprehension of the stress field within geometries akin to our case, we focus on the main rupture patterns (C1, C2, and C3) and we examine three key stress-related parameters: the principal angle ($\theta_p$) which is also the angle between the direction of $\sigma_1$ and the x-axis, and thus the equivalent of $\Psi_b$ (2) for a horizontal basement, the mean stress ($p$) and the equivalent deviatoric stress ($q$) defined as follows:

$$p = \frac{\sigma_1 + \sigma_2 + \sigma_3}{3}, \tag{8}$$

$$q = \sqrt{\frac{1}{2}(\sigma_1 - \sigma_3)^2 + \frac{1}{2}(\sigma_1 - \sigma_2)^2 + \frac{1}{2}(\sigma_2 - \sigma_3)^2}. \tag{9}$$

### 4.2.1 Comparative analysis of stress distribution inside a given cluster

Since we lack prior knowledge of the expected stress data distribution within a given cluster (whether symmetrical or skewed), we performed a statistical analysis based on the mean and median for each parameter in a given cluster. The results were similar, so we present the mean, and standard deviation (SD) as seen in Figure 4. Note that the frictional interactions allowed between the model and the back wall, and the normal support feature defined at the frontal boundary, create high perturbations in all three parameters, and are thus considered irrelevant for our analysis.

The average principal stress angle is observed to be generally fluctuating between zero and $\pm$ 10°, indicating that the principal stress direction closely aligns with the x-axis (Figure 4a1). This alignment is predicted by the CCW theory where $\Psi_b$ is proportional to the basement friction. In fact, the weaker the basement friction angle compared to the bulk resistance, the smaller the value of $\Psi_b$. This holds true when comparing $\theta_p$ in all three clusters, where average values closer and higher than 10° are more dominant in C1 (where the basement friction angles are higher), less intense in C2, and restricted to the inherited fault plane in C3. At the same time, significant deviation associated with the inherited fault system causes $\theta_p$ to fluctuate between -15° to approximately 25°, as detected at the back-thrust location in C2. In this cluster, at the root of the inherited fault, stresses exceeding 200 MPa are accumulating for both $p$, and $q$, while the back-thrust surface presents an increase in the stress values, notably detectable for $q$ with values attaining 100 MPa. In general, the distribution of high deviatoric stress values ($> 150$ MPa) near the basement follows the rupture location. This is evident in C1 where higher values are retained at the back, while they spread to the front in C3.

As for the zones with near zero $\theta_p$ values, they are characterized by a lithostatic stress-state for both $p$ (Figure 4a2), and $q$ (Figure 4a3) with values linearly increasing with depth.

It is worth noting that $q$ is expected to be proportionate to $p$ since the stress state observed is the result of the optimized external load. At the onset of rupture, the model is considered in a deterministic chaos rupture state (Mary et al., 2013). The stresses in the model are maximized up to the point of obtaining a single element verifying the Coulomb rupture criterion (on all of its nodes). This also explains the zero standard deviations observed for these parameters in the ruptured zones as evident near the back-end of the models for C1, at the ramp and the back-thrust for C2, and in the front for C3.

With the exception of these ruptured zones, the SDs for both $p$ (Figure 4b2) and $q$ (Figure 4b3) present high fluctuations attaining 15%. This is the case for the frontal part of the model in C1, everywhere except the activated inherited fault and its respective back-thrust in C2, and the back part of the model for C3. In contrast, $\theta_p$ presents fluctuations up to 20° mainly located between the activated ramp and the created back-thrust in C2 (as observed in Figure 4b1).

The last result considers the stress decrease with depth as spotted for both $p$ and $q$ in C2 and C3. The location of these stress drops are directly related to the inherited fault activation and the creation of a back-thrust. They are well distinguished in C2 since the inherited fault is completely activated, and less prominent in C3 which points to the possibility of smaller residual activation despite the newly created fault system at the front.

We remind the reader that the values obtained may seem very high but they are merely the results of an optimization process through the use of realistic parameters. Nevertheless, these values remain possible in theory.

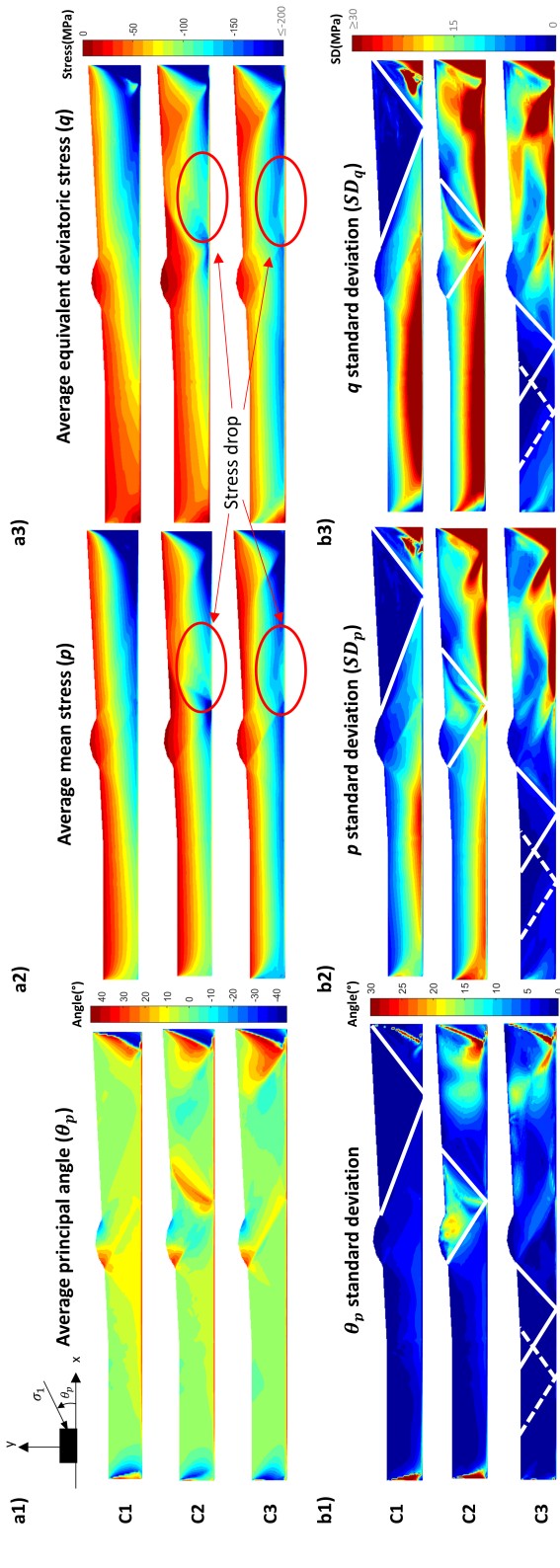

**Figure 4.** Average and standard deviation of three selected stress parameters: principal angle ($\theta_p$) between $\sigma_1$ and the horizontal x-axis, pressure ($p$), and equivalent deviatoric stress ($q$) for the three main clusters, C1, C2, and C3. The detected faults are shown in white while the stress drop zones are highlighted by red circles.

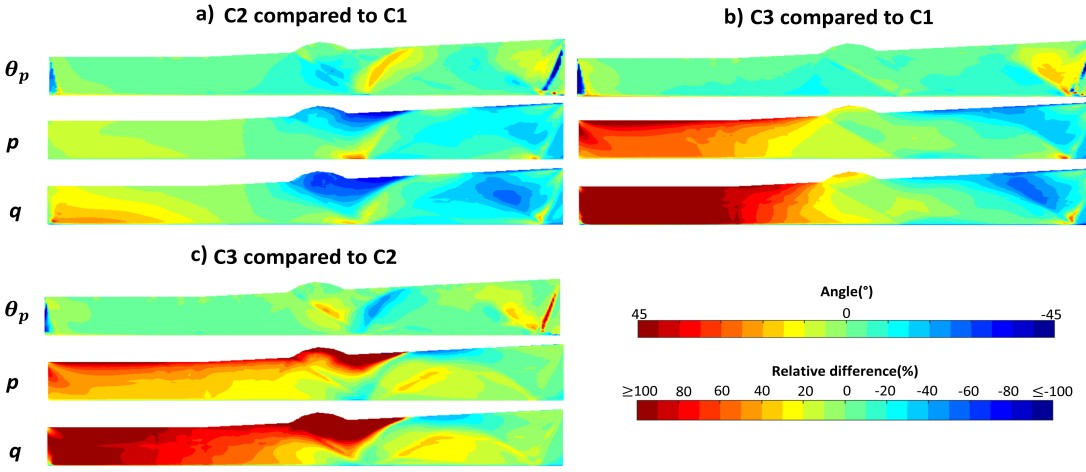

**Figure 5.** Differences in average stress fields between clusters. $C_2 - C_1$ in a), $C_3 - C_1$ in b), and $C_3 - C_2$ in c). For $\theta_p$, angle subtraction is shown (for example, $\theta_{pC_2} - \theta_{pC_1}$ in a)). For $p$ and $q$, relative difference percentages are shown (for example, $\frac{p_{C_2} - p_{C_1}}{p_{C_1}} \times 100$ in a)).

### 4.2.2 Comparative analysis of stress distribution across clusters

After analyzing the stress fields within each cluster, we conduct a comparative study to assess the differences between clusters. Each cluster is represented by its average values. When examining the differences in $\theta_p$, we observe that the higher difference values are primarily concentrated within rupture zones. For instance, when C1 is the reference cluster (Figure 5a and 5b), rupture is located at the back where the stress values are higher with near-zero deviation. Compared to C2, where the inherited ramp is activated and a back-thrust is created, angle deviations follow that of C2 since in C1 near this fault the angle fluctuation

is less than 10°, while it surpasses 25° for C2. As for the stresses, both $p$ and $q$ show negative stress differences at the back following the existence of higher stress values in these zones for C1. The same observations can be made from the comparison between C1 and C3 where the stress values for C1 are higher than their C3 counterpart at the back, resulting in negative fluctuations of up to 80%, while they are lower at the front with positive differences going beyond 100%. On the other hand, near zero stress variation is observed at the inherited fault location. As for $\theta_p$, since its values are closer to zero, following the

weaker basement friction angles in C3, the observations practically reflect the average $\theta_p$ values obtained for C1.

Finally, comparing C3 to C2 yields the same results for $\theta_p$: the angle difference reflects the same tendencies as C2 while the relative stress differences show positive values at the front, following the higher stress concentration in these locations. At the back of the model, the relative difference is less than 20%, meaning that these two clusters practically show the same stress field at the back.

The results of both statistical comparisons, suggest that higher stress values are a solid indicator of imminent failure both through the re-activation or the creation of a new faulting system, while stress rotations are more frequent near the activation of inherited faults.

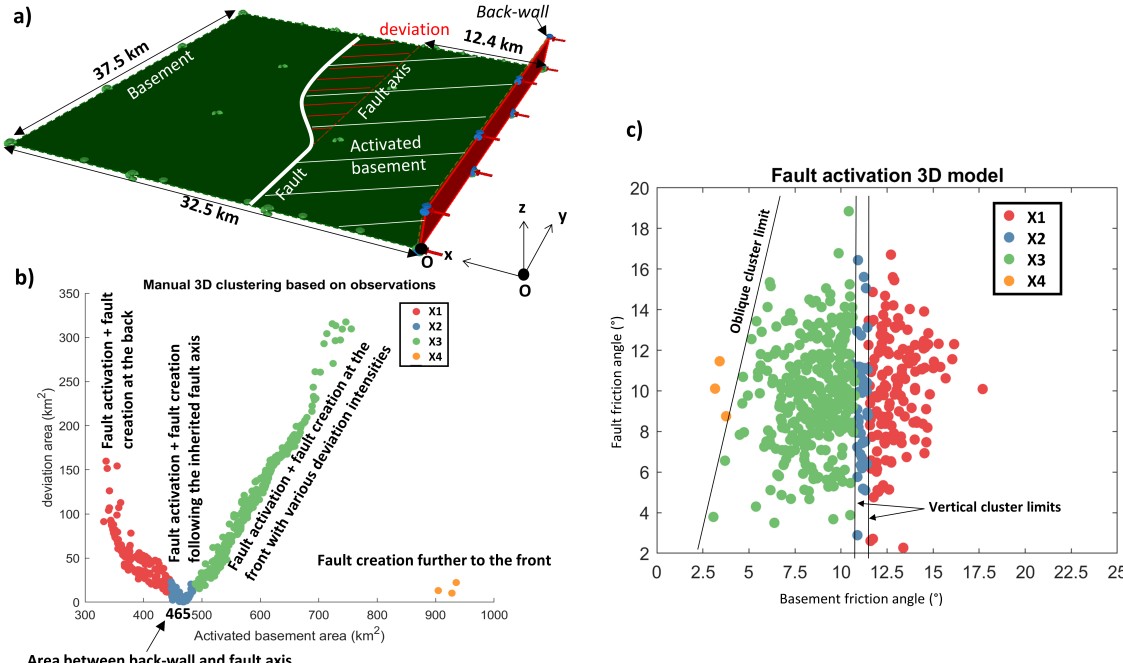

**Figure 6.** a) Defining clustering descriptors: The basement is in green, the activated part is highlighted through a pattern of parallel white lines while the deviation is highlighted by red parallel lines (the case represented is an example of cluster X3). b) Distribution of the deviation area in function of the activated basement area for each simulation. Four distinct clusters are identified and represented by different colors. c) Partition of each simulation based on both basement and fault friction angles. Each simulation is color-coded according to its respective cluster.

## 5    3D analysis

### 5.1    3D rupture pattern and clustering

We follow the suggestions of Adwan et al. (2024) and identify the critical 3D criterion as $R_{crit}$ (4). We adopt this parameter and conduct a similar analysis as for our 2D model. To focus primarily on fault behavior over that of the wedge, we constrain the friction angle variations for both fault and basement within the range of [0, 20°] (Table 1), where the inherited ramp is always activated in 2D (C2). Additionally, we set the value of $\delta$ to 0.008 and apply the polynomial fitting method to extract fault surfaces in the 500 simulations. In comparison with the calibration phase, where $\delta$ was chosen as 0.1 for $R_{crit}$ (refer to

section 4.1) and for simplification reasons, we consider a much lower $\delta$ value in order to detect the rupture zones of the ramp without considering the back-thrust. This is possible since failure is more prominent at the ramp, preceding the creation of a back-thrust. We utilize the activated basement area as our clustering descriptor. As depicted in Figure 6a, and since we expect the fault to be always activated, three distinct options emerge. The fault surface can either laterally extend without deviation following its fault axis, deviate towards the front of the model, or deviate towards the back.

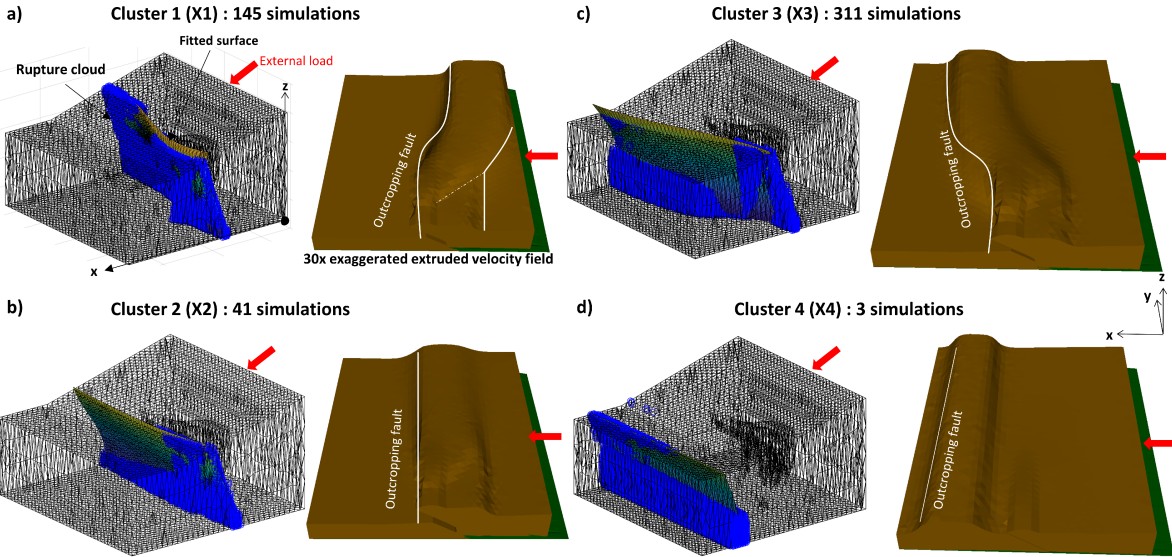

**Figure 7.** a-b-c and d) Distance-to-strain ratio $R_{crit}$ (4) verifying failure criterion and the fault surfaces fitted using polynomial fitting for the four clusters X1, X2, X3, and X4, respectively. Each point in blue refers to a node verifying the rupture criterion. An extruded velocity field is also shown to help the interpretation.

We quantify the basement fault deviation area for each simulation by assessing its intersection with the fitted surface. The extracted surface is then assimilated to a contour starting at the left end of the model (the inherited fault side) spreading to the other end and then enclosing at the same starting point x-coordinate. For example, when the fault is activated the contour closure follows the fault axis and the deviation area is calculated with respect to the existing fault plane (in red dashed lines Figure 6a). We then plot this deviation area against the basement activation area spreading between the back wall and the extracted fault. We analyze the resulting graph (Figure 6b) both manually and automatically. Following the obvious tendencies, the simulations are manually regrouped into four distinct clusters (X1, X2, X3, and X4) while the k-means method optimized with the Davies-Bouldin index further divides X3 into three distinct clusters, a classification we found excessive for this study.

Starting with lower basement activation areas, we observe a decrease in the deviation as the basement activation surface increases. This pattern identifies our first cluster (X1), characterized by the activation of the inherited fault at the left end, while at the opposing lateral end (referred to as the right end) rupture is located near the back wall (Figure 7a). The inherited fault's lateral termination, depicted through a linearly terminating blind fault, is also spotted at the surface, where a small oblique outcrop signals the creation of a deviated new fault, as is evident in the exaggerated extruded velocity field representation of Figure 7a. A total of 145 simulations exhibit this rupture pattern.

At an activation area of $465 \text{ km}^2$, the fitted surface closely aligns with the inherited fault axis (Figure 7b), leading to minimal deviation in the 41 simulations within our second cluster (X2). Furthermore, as the basement activation area increases, the deviation shifts from the back to the front of the model (Figure 7c), resulting in a deviation area increase (Figure 6b, in green).

This mechanism characterizes the 311 simulations within our third cluster (X3). Finally, in three simulations, the primary fault surface is located at the front of the model, with complete activation of the basement (Figure 7d). The newly identified contour, closes upon itself resulting in a near zero deviation area. These simulations constitute another unique cluster, labeled as X4 completely distinct from the 2D cases.

To gain further insights into these clusters, we plotted each simulation against the varying friction angles, color-coding each point according to its respective cluster (Figure 6c). A decrease in basement friction angle corresponds to a shift in fault deviation from the back end to the front end, transitioning from X1 to X4. Notably, the boundaries between these clusters are primarily vertical, emphasizing the predominant role of basement parameters in determining rupture location. The left part of the model shows a dominant inherited fault activation behavior while the right part shows the shifts in rupture location based on the basement friction angle. Any x-direction 2D cross-section taken from this part should follow the CCW theory with a critical basement angle of 9.9°. The results on the other hand prove that the shift from a back rupture to a more frontal rupture occurs at an angle closer to 11°. At the right side of the model, neither inherited faults nor geometric features are present. Two behaviors are expected: either the wedge is unstable or the basement is activated and the wedge is considered stable. The observations obtained prove a clear deviation of this norm due to the lateral interaction inside the model. The existence of an inherited fault creates a transitional phase for rupture distribution. Instead of having an abrupt shift, the inherited fault allows a progressive transition through its activation by causing the internal deformation to be localized closer to its original axis. Following the definition of the CCW theory the only stable cases are the ones obtained in X4 for basement friction angles below 4° since the bulk presents no internal deformation. This observation demonstrates that the critical basement friction angle value presents a 6° deviation between both 2D and 3D cases.

**Table 2.** Cross-sections and borehole positions examined in the 3D model (Figure 8, 9, 11).

| Cross-sections | x position | y position | z position |
|---|---|---|---|
| AA | - | 1 km | - |
| BB | - | 7 km | - |
| CC | - | 24 km | - |
| BH1 | 15 km | 1 km | - |
| BH2 | 15 km | 27 km | - |

## 5.2 3D stress analysis

We focus on X1 and X3, with prominent lateral 3D effects and abundant number of simulations and we compute both the mean and standard deviation for $p$ (Figure 8) and $q$ (Figure 9) within each cluster. We select three distinct cross-sections 'AA', 'BB', and 'CC' (Table 2 and Figure 1c) taken at different locations in our model. In 'AA', the fault is outcropping at the surface, in 'BB', it is a blind fault, while 'CC' is taken further to the right where there is no inherited fault. Both Figure 8 and 9 present the stress fields obtained over these three-cross sections in addition to the top basement surface. They follow the engineering

convention with negative stress values in compression. In what follows, we will compare and study the magnitudes of the values of $p$ and $q$ disregarding their sign in order to remove useless complications arising from different sign conventions.

### 5.2.1  Mean stress and deviatoric stresses

At the left side of the model the inherited fault is activated for both X1 and X3. Going from the back to the front of the model, high values of the mean stress $p$ are detected near the back wall and at the root of the activated fault (Figure 8a-c), while beyond the fault location $p$ is near lithostatic characterized by a linear increase with depth. Going from the left to the right of the model, the high stress concentration ($p$ values higher than 200 MPa) observed in cross-section 'AA' at the root of the activated inherited fault decreases as we follow the fault termination. This observation is validated in cross-section 'BB', where

the stress values at the root of the blind fault are less concentrated and vary between 120 MPa and 160 MPa. In these two cross sections, the mean stress decrease with depth obtained in 2D is also detectable at the foot-wall of the newly created back-thrust.

At the right side of the model, there is no inherited fault and the newly created fault system is at the back for X1 and to the front for X3. Yet, cross-section 'CC' for both clusters shows a similar stress pattern: $p$ values higher than 200 MPa are observed at the back near the basement while they decrease to 120 MPa towards the front. The only difference lies in the spread of the

values higher than 200 MPa. They are more prominent in X3 with wider distribution up to the location of the pre-existing fault (defined at the left end). Compared to the left side, high $p$ values are detected at the front for both clusters unhindered by the location of the created fault. This could be related to the essence of the LA calculation where we are on the onset of rupture and a newly created fault does not currently present any slip, preventing the convergence of stresses at its roots.

In terms of the deviatoric stress $q$ (Figure 9a-c), the observations are similar. The higher stress values obtained (above 200

MPa) are more prominent than $p$. They present a wider spread towards the front and a more important concentration. The stress decreases in the footwall of the newly created back-thrust, relative to the inherited fault activation, are also observed while the stress concentration at the root of this inherited fault are lower.

Finally, for both $p$ and $q$, the stress magnitudes present wide lateral variation as seen through cross-sections 'AA' and 'CC'. These variations can attain 40 MPa even in near-lithostatic zones proving that even in a rather small-scale area, the presence of

geological and geometrical features affect the far-field stress and creates large lateral stress variations.

In order to further understand the average stress field studied, we look at the standard deviation (SD) for both $p$ and $q$. At the front, the stress variation for X1 fluctuates between 0 MPa and 6 MPa for $p$ (Figure 8b). Higher variations are located at the left side of the model (cross-section 'AA') and disappear towards the right (cross-section 'CC'). The same observation is valid for X3 where $p$ variations attain 10 MPa in 'AA' and are near zero in 'CC'(Figure 8d). This lateral difference is also detected

in $q$ variations where at the left the standard deviation values may be greater than 15 MPa while they are less than 5 MPa at the right end of the model (Figure 9b) and (Figure 9d).

At the back, the variations are closely related to the fault location. for 'AA' and 'BB', lower variations are observed near the newly created back-thrust for $p$ and $q$ in both clusters X1 and X3. As for 'CC', low variations are observed at the back for X1 and the front for X3 following rupture location.

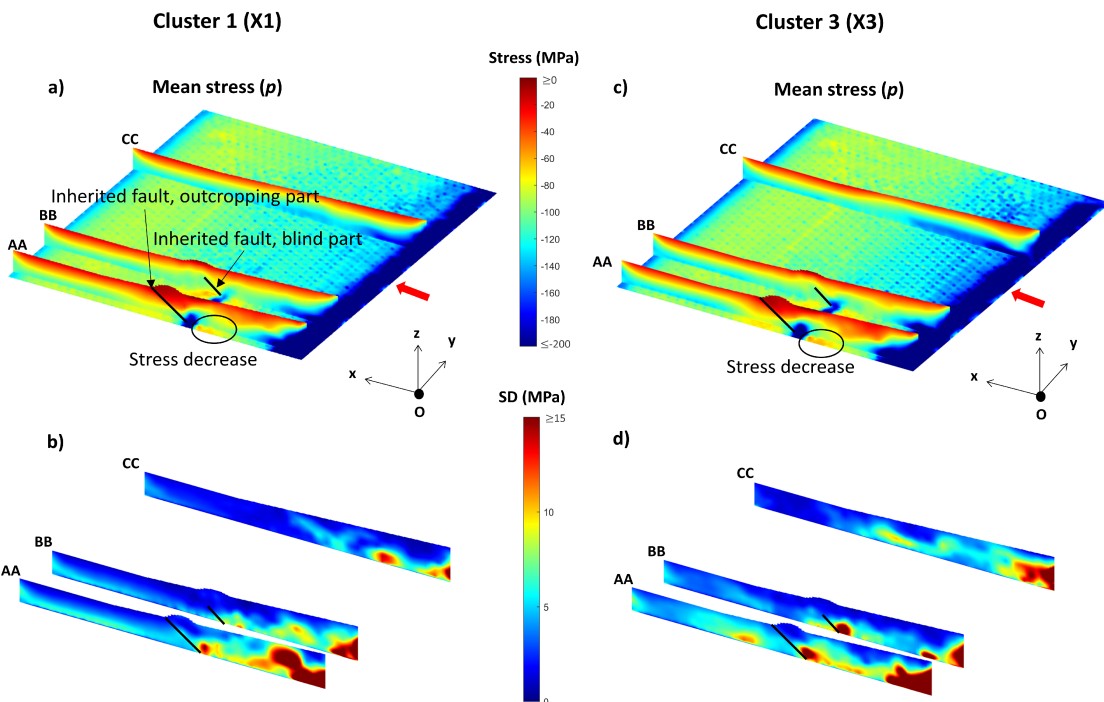

**Figure 8.** 3 different cross-sections 'AA', 'BB', 'CC', and the horizontal basement surface for clusters X1 and X3. a and c) representation of the average pressure (*p*) values respectively for X1 and X3. b and d) representations of the *p* standard deviations for X1 and X3, respectively.

### 5.2.2 Tectonic regimes

Understanding the stress state in a 3D environment also requires analyzing the principal stress directions. In contrast to its 2D counterpart, this process is more challenging. Based on the classical 'Andersonian' model of faulting (Anderson, 1951), the three primary regimes follow the Coulomb criterion and depend on which of the three principal stresses is oriented vertically. We check the direction of the three principal stresses in function of the z-axis. If $\sigma_1$ is vertical we are in a normal faulting regime, while a vertical $\sigma_2$ or $\sigma_3$ represent strike-slip or reverse faulting regimes respectively. If all principal directions are off the vertical by more than 10°, we are in a non-Andersonian regime (Hafner, 1951; Sibson, 1985; Yin and Ranalli, 1992).

We calculate the average principal directions for all simulations in a given cluster. Similarly to the 2D case, the standard deviation check showed very low direction variation between the simulations of a single cluster except near the activated inherited fault with more prominent deviations. We verify the direction of the principal stresses at each geometric node of a given tetrahedron element (Figure 10). Each element follows the dominant regime based on the directions obtained on its 4 nodes. For X1, under the applied compression load, the main model regime is reverse faulting as evident in Figure 10a, with sparse instances of strike-slips. In X3 (Figure 10b), this regime also extends to the right side of the model surface near the back wall. While the existence of such complicated stress fields near weak layers, with clear frictional contrast compared to

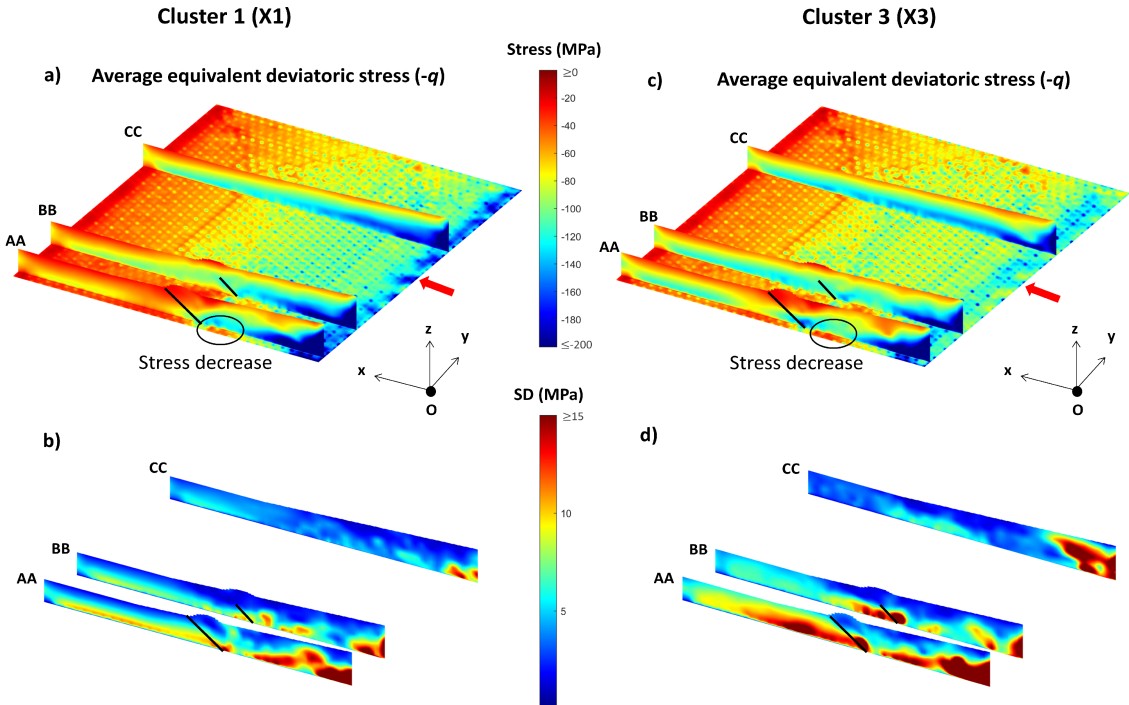

**Figure 9.** 3 different cross-sections 'AA', 'BB', 'CC', and the horizontal basement surface for clusters X1 and X3. a and c) representation of the average deviatoric stress ($q$) values for X1 and X3, respectively. b and d) representations of the $q$ standard deviations for X1 and X3, respectively.

the bulk, is explainable, their localization at the back right surface of the model is intriguing. In addition, this behavior was not
detected in the exaggerated extruded velocity field representation of Figure 7c. It might be due to surface effect caused by the creation of a back-thrust. The basement in both clusters present a complete non-Andersonian regime meaning that it undergoes a lot of stress rotation. This is also in agreement with the CCW theory where the angle $\Psi_b$ (2) is non zero due to the difference of friction between the bulk and the basement.

### 5.2.3 Simulated boreholes

Underground stress assessment primarily relies on borehole data, prompting us to conduct two borehole sections, BH1 (15 km, 1 km) and BH2 (15 km, 27 km) (Table 2 and Figure 1c). The first borehole intersects the inherited fault, while the second is positioned further to the right. We focus on evaluating $\sigma_1$, $\sigma_2$, $\sigma_3$, and $\frac{\sigma_1}{\sigma_3}$ for each borehole.

We begin with the simpler case of BH2 for both X1 and X3 (Figure 11c1-d1). As seen previously, BH2 is located in a reverse faulting zone which means that $\sigma_3$ is vertical and $\sigma_1$ is horizontal. These observations are confirmed through the clear
correlation between $\sigma_3$ and $\rho g h$, with $h$ being the depth from the surface. Conversely, $\sigma_1$ and $\sigma_2$ exhibit a "hook" shape trend, indicating a decrease in stresses toward the basement level characterized by a lower friction angle than the bulk material. At

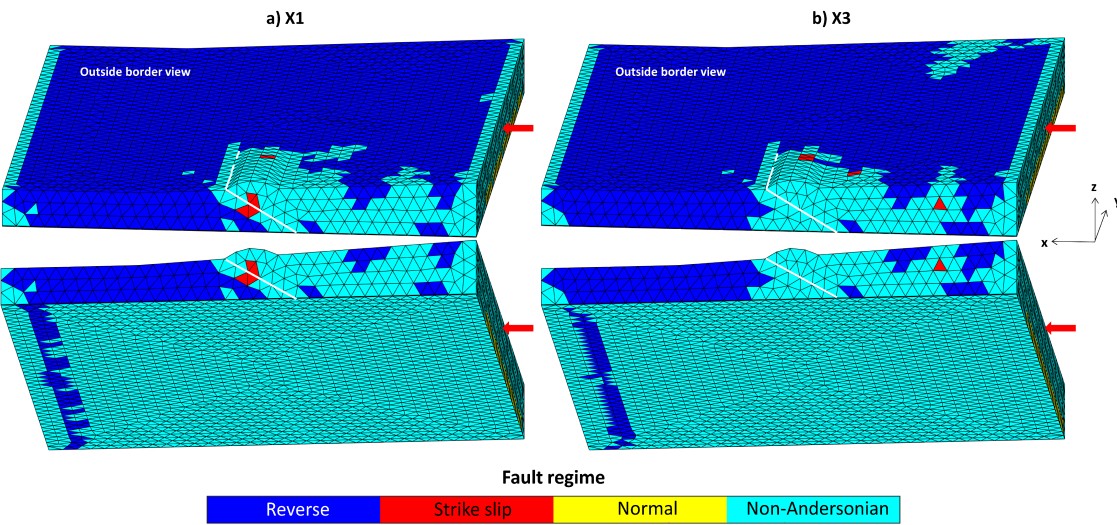

**Figure 10.** a) and b), outer representation of the fault regimes obtained respectively for clusters X1 and X3. Two different angle views are illustrated showing both the surface and the basement. Each triangular facet is colored following the dominant fault regime at the edges of its respective tetrahedron. We define the non-Andersonian state when all principal stress directions are more than 10° away from the vertical direction and we highlight the inherited fault in a white line.

the same time, $\sigma_1$ variation in function of depth is the same for both X1 and X3, which explains the conformity of the obtained $\frac{\sigma_1}{\sigma_3}$ (Figure 11c2-d2). For both clusters, the different simulations showed no variation of this ratio in BH2 presenting a clear pattern: a somewhat constant value with depth (close to 2) and an infinite horizontal asymptotic tendency at the surface where $\sigma_3$ tends to zero. The tectonic regime obtained from this borehole is reverse faulting, which is in accordance to our previous observations. Lastly, $\sigma_2$ variation in function of depth presents a clear divergence between X1 and X3 specifically beyond a depth of 1.5 km where the decrease in $\sigma_2$ values for X3 is more abrupt than that of X1 with larger variations between the simulations of a given cluster. This observation is quite interesting since it proves that by simply looking at $\sigma_1$ and $\sigma_3$, we are not able to determine any difference between X1 and X3. It is only by comparing $\sigma_2$ that the difference between these two clusters is detected. This also proves that the fault creation at the right side of this model is dependent on the lateral stress, represented by $\sigma_2$.

Now, we look at the more complex situation of BH1. Figure 11a1-b1 shows a near strike-slip tectonic regime. In this case, $\sigma_2$ is closer to $\sigma_v$ than $\sigma_3$ even though it does not completely align with it. This distance to $\sigma_v$ also explains and clarifies the previously observed non-Andersonian regime (Figure 10). At this location $\sigma_2$ orientation is closer to the z-axis but presents a deviation higher than 10°. $\sigma_1$ retains its "hook" shape tendency with depth but also shows a clear magnitude difference between X1 and X3 where $\sigma_1$ values can attain 200 MPa for X3 simulations and are capped at 180 MPa for X1 simulations. As for $\sigma_3$, from the surface to the fault location, $\sigma_3$ values are positive and near constant representing a tensile tendency. For a depth beyond the fault intersection these values shift to negative signs. This change in sign is visible in $\frac{\sigma_1}{\sigma_3}$ variation graph

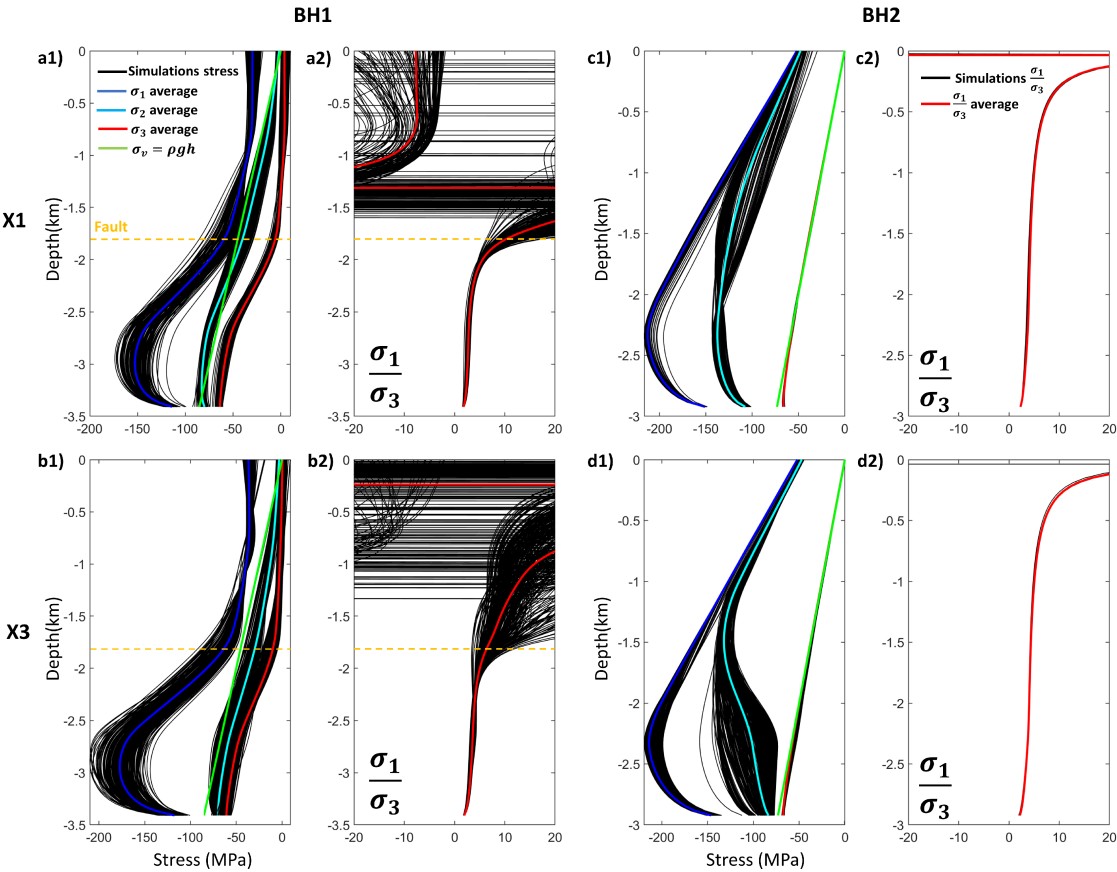

**Figure 11.** BH1: a1) and b1), BH2: c1) and d1) $\sigma_1$, $\sigma_2$ and $\sigma_3$ variation with depth for X1 and X3 respectively. The values for each simulation within a cluster are represented in black, the vertical load $\sigma_v$ value is illustrated in green, the depth of the inherited fault (if present) is denoted by an orange dashed line, and the calculated cluster average for each stress parameter under study is highlighted in blue for $\sigma_1$, cyan for $\sigma_2$ and red for $\sigma_3$. BH1: a2) and b2), BH2: c2) and d2) $\frac{\sigma_1}{\sigma_3}$ variation with depth for X1 and X3 respectively. The values for each simulation within a cluster are represented in black and the average $\frac{\sigma_1}{\sigma_3}$ value is in red.

(Figure 11a2-b2) where perturbations arise above the fault intersection. The variations between simulations of a same cluster are higher, and a clear transition from a positive ratio to a negative ratio is spotted. These observations illustrate the complexity of the stress field surrounding an activated fault surface.

## 6  Discussion

From a 3D geological perspective, studying a fault cored anticline offers direct insights into the stress field sustained by rocks during the folding process. The clustering results demonstrate that lateral stress distribution significantly alters rupture mechanisms, something traditional 2D models fail to capture.

### 6.1  2D and 3D rupture mechanisms

From a 2D perspective, the clustering results align with our expectations. An exhaustive analytic study of a perfect triangular wedge can be found in Dahlen et al. (1984). Other studies, such as Cubas et al. (2008), extended this theory to cases involving additional perturbations, such as triangular reliefs. These studies revealed a strong correlation between geometric factors and the location of thrusts. This rationale guided our modeling approach, allowing us to qualitatively confirm our method by comparing the rupture patterns we obtained with established knowledge.

In the 2D model, for fault friction angles below 20°, the activation of the inherited fault is the sole possibility, regardless of the basement friction parameters. Conversely, in the 3D models within the same friction angle range, two distinct instances of 3D lateral effects become apparent:

- The first instance is observed in cluster X4, where the basement is fully activated, and rupture occurs at the front while the inherited fault remains inactive. This cluster underscores the impact of the right part of the model (devoid of inherited faults) on the left part. Notably, failure at the front precedes and hinders the activation of the inherited fault by laterally spreading towards the left.

- The second 3D effect involves an inverse influence, where the presence of the activated inherited fault affects the CCW critical basement friction angle value. As shown in Figure 6c, the right part of the model transitions from an unstable state to a stable state for basement friction angle values below 4°, instead of 9.9°. This right-side wedge presents a wide transitional state spanning from a basement friction angle of 4° to 11.5°. In this interval, the wedge is in a critical state, with internal deformations potentially localized anywhere from the back towards the front.

This lateral 3D effect remains undetectable in 2D modeling, leading to biased interpretations and incorrect site investigation assessments. Although we focus on large parametric variations, the cluster boundaries observed in the friction angles domain prove that even small variations can lead to drastically different rupture patterns if they trigger sudden shifts in the geometry. This confirms the existence of critical behaviors in 3D heterogeneous structures beyond those portrayed by the CCW theory.

## 6.2 2D and 3D stress states

In terms of stress direction, both the 2D and 3D models exhibit a consistent stress pattern within each cluster, evident from the low standard deviation values (less than 10°) from the stress direction analysis. However, high variations are observed near the activated inherited fault, where this consistency is not maintained.

Concerning the pressure and the deviatoric stresses, the variations are more pronounced in the 2D models compared to the 3D models. Despite these differences, the stress variations in the rupture zones are minimal, nearing zero, as seen in the back for clusters C1 and X1 and at the front for clusters C3 and X3. These observations suggest that a given cluster, defined by a single rupture mechanism, can be characterized by a stress field with well-determined directions and potential variations in stress magnitudes.

Furthermore, our findings reveal stress concentrations spreading from the back wall to the zones of activated or created faults, particularly in proximity to the model's basement. High stress concentrations at the base of the inherited fault, indicate its reactivation. In contrast, in the right part of the model, where no inherited fault is present, the stress field at the onset of rupture shows no such concentrations at incipient fault roots. This observation is in accordance with the findings of Zhang et al. (2023) where the existence of weaker elements, such as inherited faults, created zones under strong compression conditions, and zones under weak compression conditions which also affected the stress concentration and propagation. But the absence of high stress values at incipient fault roots implies that in seismically active regions, predicting the formation of new fault surfaces based solely on zones of high stress concentration is unreliable. While stress magnitudes are higher, they disperse throughout the basement, intensifying only within a specific area as the major fault surface creation becomes imminent. This also validates the importance of the adopted $R_{crit}$, criterion linking both stresses and strains, in identifying these newly created fault surfaces by considering both the stress field and the deformation field representing the damage area surrounding the main fault surface.

In addition, several stress anomalies are identified. A decrease in stress values with increasing depth is observed at the foot-wall of the newly created back-thrust. Similarly, we detect tension zones at the surface of the model near the activated inherited fault similar to observations in BH1 stress logs where $\sigma_3$ shifts to positive values disturbing the $\frac{\sigma_1}{\sigma_3}$ variation with depth. These anomalies may be related to phenomena resulting from sliding caused by fault activation or creation, necessitating further investigation for adequate interpretation.

Lastly, shifts in stress direction between different tectonic regimes are observed in the 3D models. Although the chosen boundary conditions primarily lead to a dominant reverse faulting regime, stress rotations near the inherited fault caused the appearance of non-Andersonian states. These states, closer to strike-slip than reverse faulting, can be considered transitional since $\sigma_2$ is the closest to the vertical direction. This diversity in stress direction is a common observation in structurally complex zones, such as fold and thrust belts as discussed by Tavani et al. (2015). In contrast, despite BH2 being further from the inherited fault and presenting standard stress profile tendencies, a clear disruption in $\sigma_2$ is evident. This disruption is the only distinction between clusters X1 and X3, despite having completely different rupture patterns. This suggests that focusing solely on the

major and minor principal stresses or their ratio may lead to biased interpretations, as significant information depends on the lateral direction, in this case $\sigma_2$.

### 6.3   Automatic fault detection and extraction

We applied the automatic fault detection and extraction method developed by Adwan et al. (2024). For 3D, as evident in Figure 7b-c, at the right side of the model, the ruptured data cloud is closer to the detachment. This observation proves that at the onset of rupture, despite defining a pristine medium, the fault surface isn't created instantly. It starts with a series of mini-fractures at the basement level and spreads towards the surface. Under a given Cauchy distribution scale parameter $\delta$, the deeper part of the failure zone is closer to rupture than the rest of this zone, which aligns with the results obtained by Adwan et al. (2024).

### 6.4   Real world scenario implication

While the models presented in this study rely on LA calculations and involve simplified assumptions, such as homogeneous materials and idealized fault geometries, the insights derived can still be translated to real-world fault systems, particularly in complex geological settings. For example, regions like the Zagros Fold-Thrust Belt (Sepehr and Cosgrove, 2004, 2005), and the Longmen Shan range front (Burchfiel et al., 1995; Sun et al., 2022), exhibit behaviors consistent with the lateral stress effects and fault interactions observed in our 3D models. In these natural systems, lateral stress redistribution often leads to the reactivation of pre-existing faults and the formation of new faults, behaviors that our model effectively captures. This tendency was also observed in the Chi-Chi Earthquake (Taiwan) along the Chelungpu fault, part of an active thrust belt. The fault rupture propagated along a pre-existing fault, but significant lateral variations in stress were observed. After the main earthquake, lateral stress redistribution led to the activation of secondary faults and deformation in adjacent regions, altering the faulting pattern. This spread of faulting influenced the creation of new faults in zones previously thought to be stable (Ma et al., 2006). In addition, the finding that stress concentrations and rupture mechanisms are influenced by even small parametric variations emphasizes the need for site-specific analysis in real-world scenarios, where heterogeneities in material properties and geometrical complexities can lead to significant deviations from idealized predictions. Despite the synthetic nature of our models, the clustering of rupture patterns and stress field distributions provides a robust framework for understanding fault propagation, which can be applied to seismic risk assessments and structural analysis in tectonically active regions. However, caution must be exercised when applying these results directly to real-world situations. Factors such as material anisotropy, pore pressure, and more complex boundary conditions, which are not accounted for in our LA approach, could alter the stress distribution and faulting patterns in natural settings.

## 7   Conclusions

In this study, we delved into the complexities of geomechanical modeling using numerical implementations of LA in both 2D and 3D settings.

We focused on varying both basal and fault friction angles. Driven by basement activation and failure propagation, we successfully validated our approach through 2500 2D simulations, categorized into eight clusters, and 500 3D simulations grouped into four clusters. Each cluster effectively illustrated the transition from an unstable to a stable state following the CCW theory. Nonetheless, this study offers more insights into the understanding of fault dynamics by incorporating lateral stress variations, which are often overlooked in 2D models. In the vicinity of the lateral termination of a reverse fault, 2D studies would predict a small number of distinct failure mechanisms around the critical state as determined by the CCW theory. However, a full 3D calculation leads to a continuity of possible failure mechanisms. As a consequence, lateral effects can cause 2D sections to switch from stable to unstable or to new intermediate failure mechanisms, an aspect that remains undetectable in a simple 2D analysis.

The advantage of this study lies in its intensive simulation capabilities. The clustering phase allowed us to perform a statistical stress field analysis, which is quite rare in this context. In both 2D and 3D analysis the zones surrounding an activated inherited faults showed large stress values and variations, but 2D variations were more pronounced in pressure and deviatoric stresses than their 3D counterparts. Our results also proved that both 2D and 3D models exhibit consistent stress patterns within a given cluster. This means that despite high variations in basal and fault frictional properties, at the onset of rupture, the response of a given site can be the same in terms of rupture mechanism and even in terms of the stress field in ruptured zones. Nonetheless, for critical frictional values, even low parametric variations can lead to completely different behaviors. This observation highlights the importance of the CCW theory and the need to expand it into the 3D realm.

Furthermore, predicting new fault formations solely based on high stress concentrations proved unreliable due to high stress dispersion in the basement. Additionally, stress anomalies and shifts in stress direction were observed especially near the activated inherited faults, highlighting the need to consider lateral stress directions for more accurate interpretations, as significant information can be missed when focusing solely on principal stresses or their ratios.

Finally, while the models studied here are synthetic and limited in complexity, the workflow presented offers the possibility to analyze stress and deformation data applicable to real site investigations. Despite numerical limitations, interpreting the behavior of a given site at the onset of rupture provides a clear understanding of the expected failure pattern and the critical zones to avoid.

*Code and data availability.* All codes and raw data can be provided by the corresponding authors upon request.

*Author contributions.* AA, BM, PS, CB, CN, MR and TVS planned, discussed, and organized the manuscript; AA coded and performed the calculations; AA, BM, PS, CB analyzed and interpreted the data; AA wrote the manuscript draft; BM, PS, CB, CN, MR reviewed and edited the manuscript.

530 *Competing interests.* The contact author has declared that none of the authors has any competing interests

*Acknowledgements.* We are grateful to the Swiss Federal Nuclear Safety Inspectorate (ENSI), the Swiss Federal Office of Topography (swisstopo) and to CY Cergy-Paris University for financing the PhD scholarship of Anthony Adwan. We would also like to thank them for allowing us to perform pure research without any prior constraints. Finally, we also acknowledge the inspiring discussions with Yves Leroy and Guillaume Caumon.

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
