# Peer review of "Understanding the stress field at the lateral termination of a thrust fold using generic geomechanical models and clustering methods"

_EGUsphere, 2024_

## Referee Comment (RC1)

Review

Understanding the stress field at the lateral termination of a thrust fold using generic geomechanical models and clustering methods

by A. Adwan et al.

Dear Editor, dear authors,

first of all, a brief summary of the manuscript (as I understood it):

The study investigates by means of a numerical model the reactivation of a thrust fault above a detachment horizon in a compressive tectonic regime as well as the expected geometry of a lateral continuation of that thrust fault, depending on the friction of the pre-existing thrust fault and of the detachment.

With ongoing tectonic loading the Coulomb criterion is reached at some point in the model domain. At this stage the state of stress and strain in the model domain is related to certain measures quantifying the proximity to failure of the rock. The spatial distribution of these measures is used to predict at what locations a fault is likely to develop in the lateral extent of the inherited fault. It turns out that the locations of developing faults can be quite different depending on the friction angles of the detachment horizon and the inherited fault. The pairs of friction angles (of the basement surface and inherited fault, respectively) cause significant changes in the medium principal stress (minimum horizontal stress) whereas the maximum and minimum principal stresses (maximum horizontal and vertical stress) are not much affected by the frictional properties. A large amount of friction angle pairs  are tested and grouped into clusters with respect to the location of the emerging fault and reactivation of the pre-existing one.

I think the selected topic of the expectable location and shape of a laterally continued thrust fault along strike of a pre-existing fault is relevant. The manuscript is well written. Overall, I find the study worthwile for publication but there are some points that need clarification regarding the model set up and analysis. Although a generic model is presented it would be valuable to have a discussion on whether and/or how the findings of the study relate to observations.

Below you find comments refering to specific lines. The comments are grouped into more fundamental issues and small ones.

**Main comments**

Lines 65,66 and Table 1: why is only density, cohesion and internal friction angle specified for the rock mass? As I understand, the Coulomb criterion is used to define critical stress states but below this threshold there is elastic deformation involved which would require the specification of elastic parameters such as $K$, $\mu$ or $E$, $\nu$.

Line 71: it would be helpful to mention what kind of software it is: what method (finite element method?) is used to solve which equations?

Line 74: Why is the load unknown? There is no numerical solution if boundary conditions are lacking.

Line 90: lower bound limit analysis: what is that?

Line 91, 97: triangle and tetrahedral elements are those elements with the lowest numerical precision. And the number of elements used seems rather small to me and the mesh in Fig 10 looks rather coarse. Did you perform a test in order to evaluate whether the specified resolution is sufficient for reasonably small numerical errors? I see that you run quite a large number of models which would take a lot of time with high numerical resolution. But you could test one model with smaller elements for one specific set of friction angles for the fault and the detachment and compare the result with the model you have. Or even simpler, use the tetrahedron mesh and switch to second order elements if available and compare the result with the result from the model with linear elements.

Line 97: mixed bound analysis. I asume that it is explained in the given reference, but it would be helpful to briefly explain what it is. Is the reason why you use mixed bound analysis here and lower bound analysis in 2D (Line 90) the dimension (2D vs 3D)?

Line 109: ok for the 2D case but does it also hold for the 3D case if S1 is not parallel to the x axis (which is likeley the case in some areas of the 3D model)?

Line 126: I'm missing definitions of $\phi$ and c

Line 186 and 195 and caption of Fig. 4: is it the angle between S1 and the x axis or is it the angle between S1 and the x-y plane (S1 is not necessarily in the x-z-plane in the 3D case)?

Line 207/208: optimized external load: 1) what is the amount of the load? 2) with respect to what is the load optimized? Is it what you write in lines 209/210? If so, I would mention this already in the section where you describe the model set up.

Line 244: Why do you choose the value of 0.008? Do you expect your results to change qualitatively if this value is changed? Why do you choose another value than in 2D, compare Line 139? In Line 245 you call 0.008 a low value but it is four times as high as the one you apply in 2D.

Line 258: It took me quite a while to realise that the geometry of the created fault on the right side is an outcome of your simulation (or subsequent analysis). When looking back there were some hints on that (e.g. Line 49, Fig. 1c, Line 248 „three distinct options emerge", Line 278, etc.). Maybe, this is just my slow understanding but it may be helpful to others to make this more clear in advance in the introduction.

Line 262, caption of Fig. 7, Line 339 etc: velocity field: were does the velocity come from? There is no time-dependent material law implemented. Is there a time-dependent load applied at the back edge of the model? Or do you mean displacement field?

Lines 284/285: Why is there a deviation of the critical basement angle between 2D and 3D models? Is it because of different values for $\delta$? See also comment to Lines 379-381.

Line 372: I find the term „rupture" misleading. Do you actually show a rupture distribution (in terms of a length unity)? What you are showing is a dimensionless quantity related to the distance of stress and/or strain to a failure criterion. Maybe I still didn't get it. Is there actually frictional sliding and/or plastic straining of the bulk material occurring in your model or do you have basically a model with elastic bulk material in which simulation terminates as soon as the yield criterion is reached at some point? If the former is the case you could show the amount of frictional sliding and plastic strain and if the latter is the case you could plainly state it.

Lines 379-381: Here you discuss some differences in the results between your 2D model and the 3D model, which you refer to the fact that the 3D model exhibits the inherited fault only at the left side. Differences between the 2D and 3D models may also arise from the way you defined your 2D model.

Did you use plain strain, plane stress or generalised plain strain in your 2D model? See also comment to Lines Lines 284/285.

What is the reason at all why you use a 2D model?

**Small things**

Line 3. I would omit the expression „in a kilometric-scale model" at this position or shift it to the end of the sentence in order to avoid strange associations („a model developed over a basal detachment").

In line 3 it is called a „fault", otherwise it is called a „fold". I'm not a geologist but In my view what you are dealing with is a fault, not a fold. A fold may develop obove a fault but what should be folded here, the material is hnomogenous, not layered? I doubt whether it is possible to explain the creation of a fold by critical wedge theory.

Lines 9, 36, 47, 121 etc.: I find the term „rupture" misleading in this context. With rupturing I associate specifically the dynamic frictional sliding during an earthquake, which is not modelled. More general, terms like „plastic deformation", „shearing", „creation of shear planes" include aseismic behaviour as well.

Lines 20/21 and throughout the text: in general there are no brackets around year numbers if they appear within another bracket: (Zoback, 1992; Seggal and Fitzgerald, 1998; …)

Lines 29-31: „the focus was centered on … the maximum horizontal stress": yes and no. Regarding the orientation of the maximum horizontal stress, you are right. Regarding the magnitudes: in the paragraph above all those methods used to a measure stress magnitudes, particularly the more reliable ones such as hydraulic fraturing, FITs, LOTs, etc. , aim at the minimum principal stress which is never the maximum horizontal stress.

Line 65: I would call „specific gravity" gravitional acceleration

Line 66: just to be precise: as I understand, the term „basement" does not refer to the crystalline rock in contrast to overlying sediments but to the base of the model (which may nevertheless repereсent the interface between sediments above and basement rocks below) and which is concurrently implemented as a detachment horizon?

Lines 65,66: is a specific rock type inteded, the specified density may represent granite …?

Line 75: why is vertical movement not allowed? If vertical movement would be allowed the high stresses at the back may become lower as uplift is possible. … Ah  ok, you have a contact surface between the back wall and the bulk material. Is the surface of the model (except at the back edge) uncontrained?

Line 83,84: There are different mechanisms that can result in a detachment horizon e.g. layers containing salt , evaporites which would cause some sort of creeping instead of frictional sliding. Layers with very high pore pressure may result in the same phenomenon. But I agree, in terms of stress evolution above the detachment horizon there might be not so much difference depending on whether it's frictional sliding or viscous creep.

Lines 107-109: it may be helpful to have a small sketch illustrating the angles

Line 139: Is there a reason why you chose right these numbers? Do you expect qualitative changes of your results if other numbers were chosen? What geomechanical significance do these quantities have?

Line 164: in Fig. 2b you state „single V", whereas in the text you mention „two clearly distinct V positions". Is this an error or if not how can I understand that? Is it that in each model run only one V emerges but this one V occurs at two separated locations for cluster C1?

Line 160-182: I would refer several times to Fig. 3

Fig. 3 I find it a little difficult to discriminate the green lines from the colours of the contour plot. How about replacing the thick green lines by thin dashed white lines?

The contour colour is T, but where is d mentioned in the caption? Is d shown in terms of T (eq. 7)?

Line 197: replace that by than

Line 201: stresses exceeding 200 MPa at relatively shallow depth is quite a lot. Considering that these values are not principal stress magnitudes but mean stress and deviatoric stress, respectively, some principal stress magnitudes must be even way higher (e.g. SV ~ 95 MPa at most in your model, when calulated as the weight of the overburden). Although this research is an academic exercise I wonder if such high stresses are in any way realistic in nature.

Line 216, also Lines 11/12: stress decrease with depth: I only see an increase of stress with depth. Please describe, where you see it. Or do you mean the lateral decrease of stress which you refer to as stress drops in the following sentence? I would not call these stress drops as this term is often associated with the stress release occurring during an earthquake (see also Fig. 4 and caption etc.).

Fig. 10: I think most readers will know what mean stress and deviatoric stress is but it may be helpful to some if a definition is given (both in 2D and 3D).

In the caption of Fig. 4 I would replace the first word „Mean" by „Average" as indicated in the figure to avoid confusion with mean stress.

Line 236: I would be more specific: not necessarily high stress values in general indicate imminent failure but rather high deviatoric stresses. If the stress state is isotropic high stress levels should not lead to failure since the compressive strength of rocks is generally much higher than shear strength. Also the term „high" per se is not necessarily indicative of failure but high in relation to the yield level which increases with mean stress.

Caption of Figure 6 and/or Line 249: I would mention that in Fig. 6a one example of the cluster X1 is shown.

Line 257:  insert friction between basement and angle

Lines 291/292: why not following the sign convention used in geoscience? This would make some of your descriptions easier to understand, at least it should be consistent. E.g. in Fig. 8 you write „stress decrease" but with the engineering convention it would be an increase. In Line 302 you write „values higher than 200 MPa" but with the engineering convention it would be smaller than -200 MPa, Line 348: decrease etc.

Fig. 8: Just suggestions:

In the scale for stress blue colour mean high stress whereas in the scale for standard deviation red colour mean high stress. I would find it easier  if the same convention was used

It may be helpful to draw lines in the four figures that indicate the course oft he fault, which is different in a/b and c/d

Caption of Fig. 8: it should be a and c instead of a-c and it should be b and d instead of c-d

Lines 308/309: Again, the deviatoric stresses are very very high, see comment above. What do you mean by „prominent" in Line 309? At the basement it seems that q is smaller than p (compare Figs. 8 and 9) or do you mean variability?

Caption of Fig. 9: it should be a and c instead of a-c and it should be b and d instead of c-d

Line 315: I wonder if „validate" is the correct word here. If you validate a result you would need some independent information that is not part of the model.

Line 324: „rupture location prediction" sounds like forecasting the hypocentre of a future earthquake. Is it that what you intend?

Line 328: principal instead of principle

Fig. 10: There is hardly any difference between X1 and X3 and the information content of this contour plot is limited. I would suggest to use not the Andersionian classification of tectonic regimes but a scale of the tectonic regime as suggested by Simpson (1997):

Simpson R.W., 1997. Quantifying Anderson's fault types, J. Geophys. Res., 102, 17909–17919. https://doi.org/10.1029/97JB01274

Line 346: I would find SH/SV more helpful than S1/S3 because interpreting S1/S3 involves dealing with the orientation of the principal stress axes.

Line 351/352: „constant value with depth (equal to 2)": In the figures S1/S3 seems to be higher than 2 and not constant over the depth range of the model

Line 364: The tensional stresses of S3 extend to depths of ~1.2 km. Are you aware of observations of tensional stresses at such depths?

Line 366/367: Are you referring to Fig. 11a2 and b2? I do not see what you mean. S1/S3 = 2 and turning to negative values …

Fig. 11: At the surface S1 approaches or even exceeds 50 MPa, which is quite a lot. Do you know of any observations of such high stresses at the surface? Of course, your model is generic but nevertheless it should be made clear in what aspects the model explains realistic conditions and discuss potential simplifications that cause the model results to deviate from observations (e.g. weathered fractured zone in proximity to the surface with lower stiffness and/or strength, etc.)

Line 377: I think validation requires a 1:1 reproduction of existing models or analytic solutions. In a less strict sense validation could mean a comparison of model results with observational data. Otherwiese I would not use the word validation but rather terms like „… qualitatively agrees with …", or discuss specific aspects of your results which are supported by previous findings.

Lines 427-429: You conclude that focusing on S1 and S3 may lead to biased interpretations. Thinking about a practical application I wonder if one is at all in a position to make interpretations based on S1 and S3. In a tectonic regime of compression what you have is basically S3 as the vertical stress from the load of the overburden. If you do stress measurements using the most common and reliable methods you get S3 which unfortunately corresponds again to SV in a compressional tectonic regime. SH=S1 is generally derived from S3 measurements with generally large uncertainties.

Line 432-434: You describe a temporal evolution („onset", „not instantly", „starts", „spreads"). If there is no time-dependent material law involved or a time-dependent load at the back I don't see where a temporal evolution should come from.

Line 515: The initial K. of the name appears twice.

A final comment:

Noticing that ENSI is among the co-authors curiosity is awakened as to what purpose this study may have in the context of the geological situation in northern Switzerland where the designated location of the Swiss repository for nuclear waste is investigated. I understand that this study, due to its generic character, is not intended and not suitable to make a contribution to the task of planning of the repository but it would be interesting to know in what aspects this study is addressing the geology/tectonics of this area (eastern prolongation of the Jura fold and thrust belt?), if at all, and in what way the model could be adjusted to consider more details of the conditions encountered there (e.g. presence of the Baden-Irchel-Herdern Lineament as a potential reactivated feature to the east (right side) of the inherited fault (Jura fold and thrust belt), changing frictional conditions of the basement from the west to the east side (left to right side) due to the thinning out of evaporitic layers which may explain the location of the present eastern end of the Jura fold and thrust belt, etc.). But I understand if you would not go into this in the manuscript and I do not expect this.

---

## Author Comment (AC2)

**Detailed Response Referee 2**

Thank you to the authors for their interesting study. However, I have some concerns regarding the assumptions and methodology. Without clarification, it is difficult for me to discuss the results. I have outlined my concerns as follows:

**1- Please clearly state the application of the current study. How does it help to characterize the specific site?**

**5- What is the initial state of stress in the system? When there is a fault and wedge-shaped structures, the system has already experienced stress changes. See the following paper: "The evolution of pore pressure, stress, and physical properties during sediment accretion at subduction zones".**

**6- Line 72: How does "Optum CE" work? Which equations are considered? How does it discretize the equations? Is there any mesh refinement scheme used? Please provide a summarized explanation.**

**7- Line 75: I noticed that the applied load is unknown, so I assume that a fixed displacement rate was implemented. Please mention this. Also, how did you verify the stability and mesh independence of the results? I noticed that a similar mesh was used for all cases.**

**8- For 2D, 10,000 elements were used, while for 3D, 40,000 elements were considered. Are the element sizes the same in both cases? If not, on what basis are the results compared? Furthermore, the mesh dependency analysis for the 3D cases is unclear, and the stability analysis is not included. Without this information, the accuracy of the results and the impact of boundary conditions are questionable (at least for me).**

In order to answer the above questions, we give a more detailed explanation of the method used in this paper. The referee is referred to section 2: "Models setup and Limit Analysis implementation" (lines 60-134).

The section was completely reviewed and details were implemented.

Briefly:

In this paper we adopt the geotechnical software Optum G2/G3. We apply limit analysis and we perform calculations on the onset of rupture. The fact that we are at the onset of rupture means that we do not need to perform a full elastoplastic analysis and thus the initial stress state becomes irrelevant in this type of calculations.

LA is a double bounded approach. The stress field obtained from LA lower bound is not only robust (Souloumiac et al., 2010) by the fact that the calculated stress state is always on the safe side of failure but it is also mathematically compliant with the fundamental principles of equilibrium and yield conditions.

The results of the Optum CE software have been tested for both 2D and 3D models in the preliminary testing phases and the values were in accordance with the analytic results (Adwan 2023) Nonetheless, the robustness of this software has been verified throughout the years with many articles using the finite element limit analysis (FELA) method incorporated. For example, Oberhollenzer et al., 2018 explained the advantage of such a method though a comparison between Optum G2, Plaxis 2D where they studied the performances of strength reduction finite element analysis (SRFEA) with finite element limit analysis (FELA), focusing on non-associated plasticity.

Adwan et al., 2024 recently introduced an automated fault detection method mainly using FELA, and showed its applicability to usual elastoplastic analysis.

As for the meshing and the type of elements used, we conducted a thorough convergence test for all kind of possible configurations going from 5000 elements and up to 100,000 elements in a 3D model that was slightly more complex than the one considered in this paper. The parameters used assure a solid convergence and high accuracy. In Figure 1 below, we present an example of convergence tests performed throughout the preliminary testing phase:

[Figure]

Figure 1: The variation of the obtained load multiplier in function of the number of elements for both lower and upper bound analyses is shown. This load multiplier is one of the criteria used in the preliminary testing phase to determine the number of meshing elements needed to obtain acceptable results between lower and upper bound analysis. It shows that the 40000 elements adopted yield an error lower that 2.5% (Adwan 2023).

Based on our convergence tests, we were able to identify two main weak points in the direct application of LA using Optum:
- The lower bound analysis shows some instability for certain high complex cases.
- The need to run the analysis twice, for lower and upper bound results, is time consuming.
These issues are common in numerical applications of Limit Analysis (LA), which is why there was a need for a mixed bound theory (Casciaro and Cascini 1982, Zouain et al., 1993, Borges et al., 1996, Krabbenhoft et al., 2007). This approach is a compromise based on both an acceptable velocity field and a stable stress field. In the present paper we adopted that approach for 3D simulations, but we kept the lower bound approach for 2D cases.

In addition, adequate comparison with higher order elements was also performed in previous studies and thus is available in this example of referenced articles: Lyamin and Sloan a-b, 2002; Krabbenhoft et al., 2005; Krabbenhoft et al., 2007. These references explain why accurate solutions can be obtained with a moderate computational effort using low-order elements, under given conditions (for example, for the upper bound analysis, kinematically admissible discontinuities must be included between adjacent elements).

Finally, as for the question about the applicability of this method and how it helps characterize a specific site, its advantage lies in its ability to detect rupture without the need for elastic parameters. With fewer parameters and efficient optimization procedures, we can consider uncertainties in the mechanical and geometric parameters by performing sufficiently numerous simulations to define categories describing the different rupture patterns. These categories can be used as reference in order to compare existing data and validate a given assumption (for example, the results will be close to a given obtained category, with a defined range of parameter values).

PS: Since in this study the geometry was not varied from a simulation to another, the uniform meshing was adopted and is valid for all the 2D cases and 3D cases.

**2- In all cases, the dip angles are constant. What would happen if these parameters were to change?**

In the preliminary testing stages of this study, we compared different inherited fault dip angles and varying fault parameters (friction angles and cohesion). The changes in the stress fields were generally very sparse, the sole difference was in the direction of the principal stresses surrounding the fault.

Yes, the change in the principal directions is important and we acknowledge that in this paper we decided to limit geometric changes and thus adopted fixed dip angles.

**3- Line 48: What do you mean by "homogeneous categories"? What is homogeneous within these categories?**

In this study, we adopted LA method (refer to answer 1), allowing us to study the onset of rupture ("Our objective is to evaluate how changes associated with the two varying parameters impact the stress field at the onset of rupture"). Since we are performing a huge number of simulations, we decided to group them based on the rupture pattern obtained. This is why we use image processing and data analysis in order to detect the failure location, including fault propagation, to group the simulations in clusters defined by these two criteria. The "homogeneous categories" refer to the obtained clusters with the same rupture pattern (number of obtained faults, their location and extension in 3D).

No corrections were implemented

**4- Line 53: How did you define the deviatoric stress?**

We follow the definition of deviatoric stress:

$$q = \sqrt{\frac{1}{2}(\sigma_1 - \sigma_3)^2 + \frac{1}{2}(\sigma_1 - \sigma_2)^2 + \frac{1}{2}(\sigma_2 - \sigma_3)^2}$$

Where $\sigma_i$ are the three principal stresses. This definition was added line 225 of the revised manuscript.

**9- Line 64: Did you evaluate the capability of the "uniform bulk Coulomb material" in modeling real cases or sandbox models within the proposed framework of this manuscript? Please include the validation results.**

We know that Mohr-Coulomb gives a realistic representation of failure in geomaterials. The referee points to capability of the "uniform bulk Coulomb material" in modeling real cases or sandbox models. All these questions are linked and well known in the structural geology community (as

shown by Krantz 1991, Schellart 200, Lohrmann et al. 2013, Maillot 2013, ...). The peak deviatoric stress are treated simply by defining two sets of Coulomb parameters: one at peak, called the peak -- or static -- values, and one achieved after a stable behavior is reached following the peak, called "stable" -- or "dynamic" --.

The difference between these values is often associated to the initial density of the material, but it also dependent on the confining pressure during the test. If the material is not very compacted and in a low confining pressure, it will not develop much dilatancy, and the peak and stable strengths will be close. Of course, since we do not treat the ensuing deformation (onset of rupture), we do not need a second set of Coulomb values.

About the uniformity... recall that we have distinct basement and fault Coulomb parameters. Only the material is uniform. Of course, a layered material could have been considered, at the cost of a more complex analysis. We think this can be postponed for a case study.

**10- Line 148: In all cases, back-thrust is observed. However, in the literature, there are instances where back-thrusting does not occur. If the entire range of parameters is explored, some cases would likely show no back-thrust. Clarification is needed here. See the following paper: "Control of décollement strength and dip on fault vergence in fold-thrust belts and accretionary prisms."**

This study focuses on the onset of rupture. Material in the hanging wall slides over the existing or created ramp/fault marking a discontinuity in the velocity field which is represented by the back thrust. As stated by Cubas et al., 2008 this back-thrust should be seen as a migrating hinge since materials from the back stop are crossing it to reach the hanging wall. This analysis follows the assumption that every material block undergoes rigid body motion. Thus, a material points from the back-stop region would be translated toward the back thrust, be sheared when crossing it, and then be translated again parallel to the ramp. Therefore, what we call back-thrust in this paper are really merely the hinges of the imposed ramp which is really a fore-thrust. So, we do agree with that remark: we clarified our definition of back-thrusts as being only the conjugate fault of a fore-thrust, at the onset. We also added that we do not explore here the general question of back-thrusting which has been investigated earlier with the LA method (Cubas et al., 2016), and with more classical numerical method (vonHagke et al., 2024).

**11- Pore pressure and overpressure development are not considered. What would happen if these parameters were included? What is the sensitivity of the conclusions to this parameter?**

It is true that the pore pressure and the overpressure developments are important aspects when studying such geo-mechanical models. Pore pressure is known for reducing effective stresses which can weaken the rock and make it more prone to failure or even reducing shear strength, making it easier for ruptures to initiate along pre-existing weaknesses or faults. As for the overpressure (higher pressure than the hydrostatic pressure at a given depth), it can have a role in reactivating existing faults by reducing the normal stress on the fault plane, thereby lowering the frictional resistance and potentially triggering slip or rupture along the fault.

We acknowledge these limitations, but in this study, we wanted to show the tendency of rupture in the context of a fault termination. The lateral propagation of the existing fault will not change. Yes, with the consideration of these parameters some cases might shift from a cluster to another since the existing fault might be activated more easily but the overall conclusion on the frontal, back or even extending propagation of the inherited fault stands.

As for the stress fields, we don't believe that the pore pressure will have a huge influence on the stress direction, yet it will alter the stress magnitudes. Nonetheless, each failure pattern will conserve the same stress distribution and orientation.

Finally, the comparison between 2D and 3D cases remains valid, despite such simplifications.

We thank the referee for these questions and we hope that our answers offered the needed information for the referee to discuss the results of this study.

PS: Following Both referees comments and questions, adjustments have been made to the manuscript in order to better explain the methodology and present LA in a more detailed way. We provide a manuscript with marked changes.

**References cited in this response:**

- Adwan, A., 2023. Analyse mécanique stochastique des structures géologiques compressives tridimensionnelles au-dessus d'un socle rigide (Doctoral dissertation, CY Cergy Paris Université).
- Adwan, A., Maillot, B., Souloumiac, P. and Barnes, C., 2024. Fault detection methods for 2D and 3D geomechanical numerical models. International Journal for Numerical and Analytical Methods in Geomechanics, 48(2), pp.607-625.
- Cubas, N., Leroy, Y. M., & Maillot, B. (2008). Prediction of thrusting sequences in accretionary wedges. Journal of Geophysical Research: Solid Earth, 113(B12).
- Cubas, N., Souloumiac, P. and Singh, S.C., 2016. Relationship link between landward vergence in accretionary prisms and tsunami generation. Geology, 44(10), pp.787-790.
- Dahlen, F. A. (1984). Noncohesive critical Coulomb wedges: An exact solution. Journal of Geophysical Research: Solid Earth, 89(B12), 10125-10133.
- Drucker, D. C., Prager, W., and Greenberg, H. J.: Extended limit design theorems for continuous media, Quarterly of applied mathematics, 9, 381–389, 1952.
- Krabbenhoft, K., Lyamin, A.V., Hjiaj, M. and Sloan, S.W., 2005. A new discontinuous upper bound limit analysis formulation. International Journal for Numerical Methods in Engineering, 63(7), pp.1069-1088.
- Krabbenhøft, K., Lyamin, A.V. and Sloan, S.W., 2007. Formulation and solution of some plasticity problems as conic programs. International Journal of Solids and Structures, 44(5), pp.1533-1549.
- Krantz, R.W., 1991. Measurements of friction coefficients and cohesion for faulting and fault reactivation in laboratory models using sand and sand mixtures. Tectonophysics, 188(1-2), pp.203-207.
- Lohrmann, J., Kukowski, N., Adam, J. and Oncken, O., 2003. The impact of analogue material properties on the geometry, kinematics, and dynamics of convergent sand wedges. Journal of Structural Geology, 25(10), pp.1691-1711.
- Lyamin, A.V. and Sloan, S.W., 2002. Upper bound limit analysis using linear finite elements and non-linear programming. International Journal for Numerical and Analytical Methods in Geomechanics, 26(2), pp.181-216.
- Lyamin, A.V. and Sloan, S.W., 2002. Lower bound limit analysis using non-linear programming. International journal for numerical methods in engineering, 55(5), pp.573-611.
- Maillot, B., 2013. A sedimentation device to produce uniform sand packs. Tectonophysics, 593, pp.85-94.

- Oberhollenzer, S., Tschuchnigg, F. and Schweiger, H.F., 2018. Finite element analyses of slope stability problems using non-associated plasticity. Journal of Rock Mechanics and Geotechnical Engineering, 10(6), pp.1091-1101.
- Salençon, J.: Théorie de la plasticité pour les applications à la mécanique des sols, Eyrolles Paris, 1974.
- Salençon, J.: Calcul à la rupture et analyse limite, Presses des Ponts et Chaussées, 1983
- Schellart, W.P., 2000. Shear test results for cohesion and friction coefficients for different granular materials: scaling implications for their usage in analogue modelling. Tectonophysics, 324(1-2), pp.1-16.
- Souloumiac, P., Krabbenhøft, K., Leroy, Y. M., & Maillot, B. (2010). Failure in accretionary wedges with the maximum strength theorem: numerical algorithm and 2D validation. Computational Geosciences, 14, 793-811.
- von Hagke, C., Bauville, A. and Chudalla, N., 2024. Control of décollement strength and dip on fault vergence in fold-thrust belts and accretionary prisms. *Tectonophysics*, *870*, p.230172.

---

## Author Comment (AC3)

**Detailed Response Referee 1**

Regarding the brief summary and the general reviewer comment:

We thank the reviewer for their constructive feedback on our manuscript.
We appreciate the recognition of the relevance of our topic and their acknowledgment of the quality of our writing.
We are pleased to hear that they consider our study worthwhile for publication.

Below, we address the specific points raised regarding the model setup, analysis, and the relevance of our findings to observations.

**Main comments:**

**Q1:** "
- Lines 65,66 and Table 1: why is only density, cohesion and internal friction angle specified for the rock mass? As I understand, the Coulomb criterion is used to define critical stress states but below this threshold there is elastic deformation involved which would require the specification of elastic parameters such as K, µ or E, ν.
- Line 71: it would be helpful to mention what kind of software it is: what method (finite element method?) is used to solve which equations?
- Line 74: Why is the load unknown? There is no numerical solution if boundary conditions are lacking.
- Line 90: lower bound limit analysis: what is that?
- Line 91, 97: triangle and tetrahedral elements are those elements with the lowest numerical precision. And the number of elements used seems rather small to me and the mesh in Fig 10 looks rather coarse. Did you perform a test in order to evaluate whether the specified resolution is sufficient for reasonably small numerical errors? I see that you run quite a large number of models which would take a lot of time with high numerical resolution. But you could test one model with smaller elements for one specific set of friction angles for the fault and the detachment and compare the result with the model you have. Or even simpler, use the tetrahedron mesh and switch to second order elements if available and compare the result with the result from the model with linear elements.
- Line 97: mixed bound analysis. I assume that it is explained in the given reference, but it would be helpful to briefly explain what it is. Is the reason why you use mixed bound analysis here and lower bound analysis in 2D (Line 90) the dimension (2D vs 3D)?
- Line 207/208: optimized external load: 1) what is the amount of the load? 2) with respect to what is the load optimized? Is it what you write in lines 209/210? If so, I would mention this already in the section where you describe the model set up.
- Line 262, caption of Fig. 7, Line 339 etc: velocity field: where does the velocity come from? There is no time-dependent material law implemented. Is there a time-dependent load applied at the back edge of the model? Or do you mean displacement field?
- Line 372: I find the term "rupture"misleading. Do you actually show a rupture distribution (in terms of a length unity)? What you are showing is a dimensionless quantity related to the distance of stress and/or strain to a failure criterion. Maybe I still didn't get it. Is there actually frictional sliding and/or plastic straining of the bulk

material occurring in your model or do you have basically a model with elastic bulk material in which simulation terminates as soon as the yield criterion is reached at some point? If the former is the case you could show the amount of frictional sliding and plastic strain and if the latter is the case you could plainly state it.

- Did you use plain strain, plane stress or generalized plain strain in your 2D model? See also comment to Lines 284/285.

„

In order to answer the above questions, we give a more detailed explanation of the method used in this paper. The referee is referred to section 2: Models setup and Limit Analysis implementation (lines 60-134).
The section was completely reviewed and details were implemented.

Briefly:

In this paper we adopt the geotechnical software Optum G2/G3. We apply limit analysis and we perform calculations on the onset of rupture. Since it's a geotechnical software, generalized plane strain are used where out-of-plane strain is typically constrained. In this paper, we follow the same principles and we define our boundary conditions as presented in lines 99- 112 (new draft). LA is a double bounded approach. The stress field obtained from LA lower bound is not only robust (Souloumiac et al., 2010) by the fact that the calculated stress state is always on the safe side of failure but it is also mathematically compliant with the fundamental principles of equilibrium and yield conditions.
The results of the Optum CE software have been tested for both 2D and 3D models in the preliminary testing phases and the values were in accordance with the analytic results (Adwan 2023) Nonetheless, the robustness of this software has been verified throughout the years with many articles using the finite element limit analysis (FELA) method incorporated. For example, Oberhollenzer et al., 2018 explained the advantage of such a method though a comparison between Optum G2, Plaxis 2D where they studied the performances of strength reduction finite element analysis (SRFEA) with finite element limit analysis (FELA), focusing on non-associated plasticity. Adwan et al., 2024 recently introduced an automated fault detection method mainly using FELA, and showed its applicability to usual elastoplastic analysis.
As for the meshing and the type of elements used, we conducted a thorough convergence test for all kind of possible configurations going from 5000 elements and up to 100,000 elements in a 3D model that was slightly more complex than the one considered in this paper. The parameters used assure a solid convergence and high accuracy. In Figure 1 below, we present an example of convergence tests performed throughout the preliminary testing phase:

[Figure]

Figure 1: The variation of the obtained load multiplier in function of the number of elements for both lower and upper bound analyses is shown. This load multiplier is one of the criteria used in the preliminary testing phase to determine the number of meshing elements needed to obtain acceptable results between lower and upper bound analysis. It shows that the 40000 elements adopted yield an error lower that 2.5% (Adwan 2023).

Based on our convergence tests, we were able to identify two main weak points in the direct application of LA using Optum:
- The lower bound analysis shows some instability for certain high complex cases.
- The need to run the analysis twice, for lower and upper bound results, is time consuming. These issues are common in numerical applications of Limit Analysis (LA), which is why there was a need for a mixed bound theory (Casciaro and Cascini 1982, Zouain et al., 1993, Borges et al., 1996, Krabbenhoft et al., 2007). This approach is a compromise based on both an acceptable velocity field and a stable stress field. In the present paper we adopted that approach for 3D simulations, but we kept the lower bound approach for 2D cases.

In addition, adequate comparison with higher order elements was also performed in previous studies and thus is available in this example of referenced articles: Lyamin and Sloan a-b, 2002; Krabbenhoft et al., 2005; Krabbenhoft et al., 2007. These references explain why accurate solutions can be obtained with a moderate computational effort using low-order elements, under given conditions (for example, for the upper bound analysis, kinematically admissible discontinuities must be included between adjacent elements).

Many thanks for your remarks!!

**Q2:** "Line 109: ok for the 2D case but does it also hold for the 3D case if S1 is not parallel to the x axis (which is likely the case in some areas of the 3D model)?"

We thank the referee for this question reformulated as follow: is the Coulomb Critical Wedge (CCW) theory valid for 3D ?

The CCW theory is only formulated in 2D. There is no definition of a 3D orientation of the principal stress. This question constitutes a research topic we are recently working on. An

article is in preparation (by Pauline Souloumiac) considering the validity of the CCW theory in 3D while also trying to find a more suited 3D formulation.

This means that one must distinguish between the CCW theory (which is only 2D), and the definition of stability of a wedge (no internal deformation), which is valid both in 2D and 3D. All the 3D cases studied here are considered unstable since faults are being created (be it at the back, front or a continuation of the existing fault axis).

Now to finalize the answer to your question, despite the difference spotted between 2D and 3D analysis, the CCW theory is valid for 3D as was also proven by our analysis and interpretations (Line 340, Line 394…).

**Q3:** "Line 126: I'm missing definitions of $\emptyset$ and c"

These terms represent the internal friction angle and the cohesion respectively.
So, if an element is located at material A (for example), the elements nodes will take the material's parameters.
Changes have been added to Section 2D analysis line 164.

**Q4:** "Line 186 and 195 and caption of Fig. 4: is it the angle between S1 and the x axis or is it the angle between S1 and the x-y plane (S1 is not necessarily in the x-z-plane in the 3D case)?

In 2D, since we only have x-y plane, the calculated principle angle is between S1 and the x-axis (direction of the applied external load as stated in line 221). Now in 3D, it is assimilated to the angle between S1 and the x-y plane (here defined as the horizontal plane). Nonetheless, in this paper, we opted for a tectonic regime analysis instead of similar stress direction analysis following $\theta$. This is why we verified the angle between each of the principal stresses and the vertical axis in order to determine the dominant regime (as defined by Anderson).

**Q5:** "
- Line 244: Why do you choose the value of 0.008? Do you expect your results to change qualitatively if this value is changed? Why do you choose another value than in 2D, compare
- Line 139? In Line 245 you call 0.008 a low value but it is four times as high as the one you apply in 2D.
"

To answer these questions, we refer the referee to the recent published article Adwan et al., 2024.
In this article, a rupture (fault, fissure…) automatic detection method is presented in detail. The author, introduces $\delta$ a rupture scale coefficient responsible for adequately determining a rupture zone. If this parameter is set to a very small value, the detected rupture zone is thin, discontinuous or even formed by just one element (3nodes in 2D and 4 in 3D). This will definitely hinder rupture detection and thus $\delta$ must be configured adequately in order to perform the automatic detection process.
In this study, we calibrated $\delta$ using 3 extreme cases and we launched the simulations. In 2D, since we are adopting the image processing scheme, $\delta$ values are less demanding but in 3D

and following the data analysis and the suggestions of Adwan et al., 2024, we calibrated $\delta$ in order to just obtain the main fault without the created back-thrust (for simplification reasons). (Small adjustments were added lines 284-290)

**Q6:** "Line 258: It took me quite a while to realize that the geometry of the created fault on the right side is an outcome of your simulation (or subsequent analysis). When looking back there were some hints on that (e.g. Line 49, Fig. 1c, Line 248 „three distinct options emerge ", Line 278, etc.). Maybe, this is just my slow understanding but it may be helpful to others to make this clearer in advance in the introduction."

We thank the referee for this question, and we confirm that small adequate changes have been made throughout the paper in order to render this part more fluid (for example line 290).

**Q7:** "

- Lines 284/285: Why is there a deviation of the critical basement angle between 2D and 3D models? Is it because of different values for $\delta$? See also comment to Lines 379-381.
- Lines 379-381: Here you discuss some differences in the results between your 2D model and the 3D model, which you refer to the fact that the 3D model exhibits the inherited fault only at the left side. Differences between the 2D and 3D models may also arise from the way you defined your 2D model.
- What is the reason at all why you use a 2D model?

"

One of the conclusions of this paper is the difference in the critical basement angle between 2D and 3D.

Following the CCW theory (refer to Q2), the unstable state is defined as the state where internal deformation exists in the wedge.

In 2D the jump between rupture at the back (touching the back-wall) and a full basement activation without internal deformation is direct and thus the parameters that cause such a change are considered as the critical parameters.

The difference is that in 3D, between rupture located at the back and a full decollement activation, there is a range of parameter that causes internal deformation in the wedge and thus creates the deviated fault planes. This phenomenon is due to the existence of an inherited fault at one end of the model (left side) while the right side remains unchanged.

The analysis showed that if we consider a given cross-section on the right side and without taking the left-side into consideration, the behavior of the wedge will follow the 2D CCW theory and thus the critical parameters are the ones obtained for a simple 2D analysis.

Now in 3D, there is a lateral effect. As was see in this paper, sometimes the left side bulk (sliding over the fault for example) pulls the right-side bulk and causes the fault creation to deviate from the back-wall (which is a case that doesn't exist in the 2D CCW theory).

And vice-versa, the right side that should have attained stability is denied this state by the left side and thus the fault is retained to a different position in the model without having any geometric capture points (Cubas et al., 2008) which is also a case not listed in the 2D theory. This aspect is one of the reasons why we also performed the 2D analysis. We wanted to validate the theory and show that ramp/back-thrusts always follow geometric points (slope termination, surface relief…) but the 3D lateral effect can cause such a system to appear in locations completely distinct from the usual 2D theory.

As for the $\delta$ factor, it has no effect on the rupture calculation. It is a binarization factor that allows us to focus on the main aspects of a rupture zone (in the case of 3D, instead of having

data points that require clustering between ramp and back-thrust, we calibrated $\delta$ to obtain the ramp without its respective back-thrust).

The discussion and conclusion were adequately adjusted in order to clarify these points.

**Small things:**

Line 3. I would omit the expression "in a kilometric-scale model" at this position or shift it to the end of the sentence in order to avoid strange associations ("a model developed over a basal detachment").
We followed the referee's suggestion and we omitted the expression

In line 3 it is called a fault, otherwise it is called a fold. I'm not a geologist but in my view what you are dealing with is a fault, not a fold. A fold may develop above a fault but what should be folded here, the material is homogenous, not layered? I doubt whether it is possible to explain the creation of a fold by critical wedge theory.
Corrections have been done in the first line of the discussion section. As for folds, we believe that even homogeneous materials can indeed form fold under external loads (sandbox experiences).

Lines 9, 36, 47, 121 etc.: I find the term "rupture" misleading in this context. With rupturing I associate specifically the dynamic frictional sliding during an earthquake, which is not modelled. More general, terms like „plastic deformation ", „shearing ", „creation of shear planes "include aseismic behavior as well.
We understand the referee's point of view, but in this article, we are applying the LA theory. In our modeling scheme, rupture occurs when the model fails to resist the external applied load. This rupture is represented by internal deformation leading to the creation of a rupture surface, in this case a fault.
We decide to maintain the definitions used in this draft.

Lines 20/21 and throughout the text: in general there are no brackets around year numbers if they appear within another bracket: (Zoback, 1992; Seggal and Fitzgerald, 1998; …)
We agree with the referee, and we corrected all the instances where such a miss-shape happened.

Lines 29-31: „the focus was centered on … the maximum horizontal stress": yes and no. Regarding the orientation of the maximum horizontal stress, you are right. Regarding the magnitudes: in the paragraph above all those methods used to a measure stress magnitudes, particularly the more reliable ones such as hydraulic fraturing, FITs, LOTs, etc. , aim at the minimum principal stress which is never the maximum horizontal stress.
We thank the referee for this remark, we changed the formulation of the phrase in order to conserve the intended meaning.

Line 65: I would call „specific gravity" gravitional acceleration
Correction was performed.

Line 66: just to be precise: as I understand, the term „basement" does not refer to the crystalline rock in contrast to overlying sediments but to the base of the model (which may nevertheless reperesent the interface between sediments above and basement rocks below) and which is concurrently implemented as a detachment horizon?
Yes, your understanding is exact. What we refer to as basement is in fact the decollement level, defined as a shear plane with varying friction angles.

Lines 65,66: is a specific rock type inteded, the specified density may represent granite …?

We do not consider a specific rock in our simulations, but we adopted parameters that are commonly used in such configurations and that can be considered realistic.

Line 75: why is vertical movement not allowed? If vertical movement would be allowed the high stresses at the back may become lower as uplift is possible. … Ah ok, you have a contact surface between the back wall and the bulk material. Is the surface of the model (except at the back edge) uncontrained?
"- At the frontal edge and on both lateral sides (for the 3D case), normal supports are defined. These supports exclusively counteract forces perpendicular to the edge planes, preventing any movement in that direction. This also means that the movements parallel to the edges are free." (Line 104)

It is exactly as the referee pointed, only the surface in contact with the applied load is constrained through the definition of a friction plane between the rigid plate and the model. We agree with the referee that, had we nullified the friction of this rigid plate, the stress concentration at the back wouldn't have caused a problem, but we wanted to replicate a more realistic model. Nonetheless, had these concentrations proven to be problematic, we would have re-launched our simulations with the adjusted parameters.

Line 83,84: There are different mechanisms that can result in a detachment horizon e.g. layers containing salt , evaporites which would cause some sort of creeping instead of frictional sliding. Layers with very high pore pressure may result in the same phenomenon. But I agree, in terms of stress evolution above the detachment horizon there might be not so much difference depending on whether it's frictional sliding or viscous creep.
No answer demanded.

Lines 107-109: it may be helpful to have a small sketch illustrating the angles
We agree with the referee about this aspect, but we wanted to minimize the figure count as much as possible and thus we thought it better to refer the reader for a more detailed explanation in the given references.

Line 139: Is there a reason why you chose right these numbers? Do you expect qualitative changes of your results if other numbers were chosen? What geomechanical significance do these quantities have?
We refer the referee to the answer given to Q5. The values chosen follow the explanation given by Adwan et al., 2024. Depending on the coefficient value chosen, more elements could be considered at rupture (nearing rupture, following the failure criterion).

Line 164: in Fig. 2b you state „single V", whereas in the text you mention „two clearly distinct V positions". Is this an error or if not how can I understand that? Is it that in each model run only one V emerges but this one V occurs at two separated locations for cluster C1?
We apologize for the misleading legend. In Fig 2b, we only show the clustering obtained for the single V cases, since it's the most straightforward result.
In each simulation, we do not know a priori how many V we will obtain. Depending on the parameters adopted, the number of faults changes and thus the number of Vs also.
In these simulations we only observed 1 or 2 Vs but in other cases 3Vs or more can be obtained or even just ramps without the back-thrusts (depending on the geometry and the $\delta$ configuration).

Line 160-182: I would refer several times to Fig. 3
Corrections where implemented.

Fig. 3 I find it a little difficult to discriminate the green lines from the colours of the contour plot. How about replacing the thick green lines by thin dashed white lines?
In the first iteration of the draft before submission, the faults were indeed represented by thick white lines, but we found it better to switch to thick green lines.

The contour colour is T, but where is d mentioned in the caption? Is d shown in terms of T (eq. 7)?
Following Adwan et al., 2024, we applied a Cauchy transformation to d and thus we obtained the transformed form where we just focus on the low values of d.

Line 197: replace that by than
Correction has been done.

Line 201: stresses exceeding 200 MPa at relatively shallow depth is quite a lot. Considering that these values are not principal stress magnitudes but mean stress and deviatoric stress, respectively, some principal stress magnitudes must be even way higher (e.g. SV ~ 95 MPa at most in your model, when calulated as the weight of the overburden). Although this research is an academic exercise I wonder if such high stresses are in any way realistic in nature.
These are the results obtained through optimization process. The values obtained are at the onset of rupture through the use of realistic Coulomb parameters. It is also worth noting that through these simulations, we do not consider any reduction in stress with time or other phenomena. Nonetheless, these results remain possible in theory.
Adequate changes have been made (line 258-259).

Line 216, also Lines 11/12: stress decrease with depth: I only see an increase of stress with depth. Please describe, where you see it. Or do you mean the lateral decrease of stress which you refer to as stress drops in the following sentence? I would not call these stress drops as this term is often associated with the stress release occurring during an earthquake (see also Fig. 4 and caption etc.).
The stress drop we are referring to is mainly visible for the deviatoric stress in C3. Near the basement in the red circle (Figure 4) there is a sort of buckling visible where the stress drops from more than 150 MPa to less than 100 MPa despite the depth.
This is what we tried to relate to the activation of the inherited fault.
Since these are literally drops in stress values, we believe that referring to them as so is acceptable.

Fig. 10: I think most readers will know what mean stress and deviatoric stress is but it may be helpful to some if a definition is given (both in 2D and 3D).
In the caption of Fig. 4 I would replace the first word „Mean" by „Average" as indicated in the figure to avoid confusion with mean stress.
Corrections have been made and the definitions of p and q were added.

Line 236: I would be more specific: not necessarily high stress values in general indicate imminent failure but rather high deviatoric stresses. If the stress state is isotropic high stress levels should not lead to failure since the compressive strength of rocks is generally much higher than shear strength. Also the term „high" per se is not necessarily indicative of failure but high in relation to the yield level which increases with mean stress.

We agree with the referee on the implication of high deviatoric stresses. In this article, we are always on the onset of rupture, so the stresses analyzed are the ones observed at failure.

Caption of Figure 6 and/or Line 249: I would mention that in Fig. 6a one example of the cluster X1 is shown.
Correction has been made.

Line 257: insert friction between basement and angle
Correction has been made.

Lines 291/292: why not following the sign convention used in geoscience? This would make some of your descriptions easier to understand, at least it should be consistent. E.g. in Fig. 8 you write „stress decrease" but with the engineering convention it would be an increase. In Line 302 you write „values higher than 200 MPa" but with the engineering convention it would be smaller than -200 MPa, Line 348: decrease etc.
We agree with the referee and we added the following sentence at the start of the 3D stress analysis: "In what follows, we will compare and study the magnitudes of the values disregarding their sign in order to remove useless complications arising from different sign conventions."

Fig. 8: Just suggestions:
In the scale for stress blue colour mean high stress whereas in the scale for standard deviation red colour mean high stress. I would find it easier if the same convention was used
It may be helpful to draw lines in the four figures that indicate the course oft he fault, which is different in a/b and c/d
We thank the referee for this suggestion. No correction was applied.

Caption of Fig. 8: it should be a and c instead of a-c and it should be b and d instead of c-d
Correction has been made.

Lines 308/309: Again, the deviatoric stresses are very very high, see comment above. What do you mean by „prominent" in Line 309? At the basement it seems that q is smaller than p (compare Figs. 8 and 9) or do you mean variability?
We refer the referee to our previous answer. As for the prominent yes we meant more distinct and with more variability.

Caption of Fig. 9: it should be a and c instead of a-c and it should be b and d instead of c-d
Adjustments have been made

Line 315: I wonder if „validate" is the correct word here. If you validate a result you would need some independent information that is not part of the model.
"validate" has been replaced by "further understand".

Line 324: „rupture location prediction" sounds like forecasting the hypocentre of a future earthquake. Is it that what you intend?
We just wanted to say following the location of the created fault whether at the back, center ot front. The word prediction was omitted.

Line 328: principal instead of principle
Correction has been made.

Fig. 10: There is hardly any difference between X1 and X3 and the information content of this contour plot is limited. I would suggest to use not the Andersionian classification of tectonic regimes but a scale of the tectonic regime as suggested by Simpson (1997):
Simpson R.W., 1997. Quantifying Anderson's fault types, J. Geophys. Res., 102, 17909–17919. https://doi.org/10.1029/97JB01274
We thank the referee for this important remark and we admit not knowing about it previously. Nonetheless, we adopted a similar approach but following limits and intervals instead of a continuous representation.

Line 346: I would find SH/SV more helpful than S1/S3 because interpreting S1/S3 involves dealing with the orientation of the principal stress axes.
We agree that adopting SH and SV would have been a better option but depending on the borehole, SH and SV can easily be related to S1 and S3. For example, looking at BH2, it's located in a reverse faulting regime zone. This means that S1 is horizontal (following the direction of the applied load) and S3 is vertical.
Adequate changes have been made.

Line 351/352: „constant value with depth (equal to 2)": In the figures S1/S3 seems to be higher than 2 and not constant over the depth range of the model
Adequate changes have been made: "Somewhat constant value (close to 2)".

Line 364: The tensional stresses of S3 extend to depths of ~1.2 km. Are you aware of observations of tensional stresses at such depths?
As stated before, these observations are the results of an optimization process on the onset of rupture. In theory they are valid and so they should be found in reality (not considering different aspects that might alter their behavior).

Line 366/367: Are you referring to Fig. 11a2 and b2? I do not see what you mean. S1/S3 = 2 and turning to negative values …
We are referring to BH1, where the ration of S1/S3 switches to negative values beyond the fault location.

Fig. 11: At the surface S1 approaches or even exceeds 50 MPa, which is quite a lot. Do you know of any observations of such high stresses at the surface? Of course, your model is generic but nevertheless it should be made clear in what aspects the model explains realistic conditions and discuss potential simplifications that cause the model results to deviate from observations (e.g. weathered fractured zone in proximity to the surface with lower stiffness and/or strength, etc.)
The values obtained are on the onset of rupture and thus they are the maximum values that can be supported by our model. It is normal for these values to be higher than the ones observed in reality.

Line 377: I think validation requires a 1:1 reproduction of existing models or analytic solutions. In a less strict sense validation could mean a comparison of model results with observational data. Otherwiese I would not use the word validation but rather terms like „… qualitatively agrees with …", or discuss specific aspects of your results which are supported by previous findings.
We agree with the referee and we adjusted the sentence as suggested.

Lines 427-429: You conclude that focusing on S1 and S3 may lead to biased interpretations. Thinking about a practical application I wonder if one is at all in a position to make interpretations based on S1 and S3. In a tectonic regime of compression what you have is basically S3 as the vertical stress from the load of the overburden. If you do stress measurements using the most common and reliable methods you get S3 which unfortunately corresponds again to SV in a compressional tectonic regime. SH=S1 is generally derived from S3 measurements with generally large uncertainties.

We agree with the referee on this point and we wanted to clarify that in general the lack of S2 refers to 2D analysis. We wanted to accentuate the fact that such analysis might give somewhat good results when compared to real observations but that is not a given.

Line 432-434: You describe a temporal evolution („onset", „not instantly", „starts", „spreads"). If there is no time-dependent material law involved or a time-dependent load at the back I don't see where a temporal evolution should come from.

We refer the referee to our answer of Q1.

Line 515: The initial K. of the name appears twice.

Adequate corrections have been made

A final comment:
Noticing that ENSI is among the co-authors curiosity is awakened as to what purpose this study may have in the context of the geological situation in northern Switzerland where the designated location of the Swiss repository for nuclear waste is investigated. I understand that this study, due to its generic character, is not intended and not suitable to make a contribution to the task of planning of the repository but it would be interesting to know in what aspects this study is addressing the geology/tectonics of this area (eastern prolongation of the Jura fold and thrust belt?), if at all, and in what way the model could be adjusted to consider more details of the conditions encountered there (e.g. presence of the Baden-Irchel-Herdern Lineament as a potential reactivated feature to the east (right side) of the inherited fault (Jura fold and thrust belt), changing frictional conditions of the basement from the west to the east side (left to right side) due to the thinning out of evaporitic layers which may explain the location of the present eastern end of the Jura fold and thrust belt, etc.). But I understand if you would not go into this in the manuscript and I do not expect this.

It is true that this article is part of the PhD of Anthony Adwan financed by ENSI, but it is also true that they allowed us to perform pure research without any Nuclear assessment concerns. Nonetheless, the methodology developed in this generic model was applied to a model inspired by the Jura setting and the results might be the subject of a later paper.

Adequate changes have been added to the acknowledgment section.

**References cited in this report:**

- Adwan, A., 2023. Analyse mécanique stochastique des structures géologiques compressives tridimensionnelles au-dessus d'un socle rigide (Doctoral dissertation, CY Cergy Paris Université).
- Adwan, A., Maillot, B., Souloumiac, P. and Barnes, C., 2024. Fault detection methods for 2D and 3D geomechanical numerical models. International Journal for Numerical and Analytical Methods in Geomechanics, 48(2), pp.607-625.
- Cubas, N., Leroy, Y. M., & Maillot, B. (2008). Prediction of thrusting sequences in accretionary wedges. Journal of Geophysical Research: Solid Earth, 113(B12).
- Drucker, D. C., Prager, W., and Greenberg, H. J.: Extended limit design theorems for continuous media, Quarterly of applied mathematics, 9, 381–389, 1952.
- Krabbenhoft, K., Lyamin, A.V., Hjiaj, M. and Sloan, S.W., 2005. A new discontinuous upper bound limit analysis formulation. International Journal for Numerical Methods in Engineering, 63(7), pp.1069-1088.
- Krabbenhøft, K., Lyamin, A.V. and Sloan, S.W., 2007. Formulation and solution of some plasticity problems as conic programs. International Journal of Solids and Structures, 44(5), pp.1533-1549.
- Lyamin, A.V. and Sloan, S.W., 2002. Upper bound limit analysis using linear finite elements and non-linear programming. International Journal for Numerical and Analytical Methods in Geomechanics, 26(2), pp.181-216.
- Lyamin, A.V. and Sloan, S.W., 2002. Lower bound limit analysis using non-linear programming. International journal for numerical methods in engineering, 55(5), pp.573-611.
- Salençon, J.: Théorie de la plasticité pour les applications à la mécanique des sols, Eyrolles Paris, 1974.
- Salençon, J.: Calcul à la rupture et analyse limite, Presses des Ponts et Chaussées, 1983
- Souloumiac, P., Krabbenhøft, K., Leroy, Y. M., & Maillot, B. (2010). Failure in accretionary wedges with the maximum strength theorem: numerical algorithm and 2D validation. Computational Geosciences, 14, 793-811.
- Zouain, N., Herskovits, J., et al. (1993). An iterative algorithm for limit analysis with nonlinear yield functions. International Journal of Solids and Structures, 30(10):1397-1417.
- Borges, L., Zouain, N., et al. (1996). A nonlinear optimization procedure for limit analysis. European Journal of Mechanics Series A Solids, 15:487-512.
- Casciaro, R. and Cascini, L. (1982). A mixed formulation and mixed finite elements for limit analysis. International Journal for Numerical Methods in Engineering, 18(2):211-243.

---

## Referee Report (RR1)

Dear Authors, dear Editor,

I find the manuscript updated according to my recommendations. Thank you for that.

Your additions helped me to understand better the method you apply. You are not modelling the stress state (the actual or expectable one) but merely the stresses you show, e.g. in Figs. 8 and 9, are stresses at which failure and plastic deformation occurs, right? This has not been clear to me before.

From my side, the manuscript is acceptable after considering some small remarks in the following:

Lines 75, 88, 98, 138 in the new version: is it really velocity or should it be displacement or strain?

Line 133/134 in the new version: I think the sentence „Beyond 100000 elements, the meshing process …" can be omitted.

Line 230 new version: I would add an s: follows

My comment to line 3 in the original version: Contrary to your statement you didn`t follow my suggestion. A model does not develop over a basal detachment.

My comment to line 20/21 in the original version: The references are modified not in all cases (e.g. Lines 22, 26, 27, 28, 30, 31, 33, etc. in the NEW version)

My comment to line 236 in the original version: what I meant was just to speak of deviatoric stress instead of stress.

Captions Figs. 8 and 9:    Still not correct,  b and d refer to SD

My comment to line Line 515 in the  original version: I don't see any changes. Comparing with the reference below it seems that it should read Krabbenhøft, K., Lyamin, A.,    instead of K. Krabbenhøft, A. L.

I just have a final remark considering the purpose and capability of models in general (I expect no action on this one). I'm starting with three citations of yours:

In line 266/267 of the new version you write: „We remind the reader that the values obtained may seem very high but they are merely the results of an optimization process through the use of realistic parameters. Nevertheless, these values remain possible in theory."

Your comment to my comment to Line 364 in the original version: „In theory they are valid and so they should be found in reality (not considering different aspects that might alter their behavior)."

Your comment to my comment on Fig. 11 in the original version: „It is normal for these values to be higher than the ones observed in reality."

I think it is valid to draw conclusions from models within the framework the model is defined. And this is what you did and have inteded. But caution is needed if conclusions on nature are made based on models. The motivation for a model of course is nature. So I find it important to draw conclusions within the model set up and then discuss in what aspects the results and conclusions may or may not reflect nature and if not what may be reasons for that (wrong asumptions, omission of relevant processes, etc.). I like the following two phrases:

"Essentially, all models are wrong, but some models are useful." (George Box)

„The purpose of models is not to fit the data but to sharpen the questions." (Samuel Karlin)

---

## Author Response (AR2)

Dear Reviewer 1, dear editor,

We would like to start by reconfirming our gratitude for the reviewer's interest in our paper. Moreover, here are the comments to the additional technical corrections:

**Lines 75, 88, 98, 138 in the new version: is it really velocity or should it be displacement or strain?**

Thank you for your insightful question regarding the use of the velocity field in the upper bound theorem. I would like to clarify the reasoning behind this choice in limit analysis.

The upper bound theorem is fundamentally a kinematic approach. In this context, the use of a velocity field is essential because the method relies on considering possible collapse mechanisms and calculating the energy dissipation rates in those mechanisms. The key is not the magnitude of displacement, but rather the rate at which work is being done and energy is being dissipated as the system moves toward failure.
Both displacement and strain fields describe the positions/deformation of a given point within the material at a particular instant, but they don't directly inform us about how fast collapse mechanisms are evolving, which is crucial for calculating the rate of energy dissipation and external work. In contrast, velocity fields describe how the material points are moving, which directly correlates to the rate of energy dissipation and external work — the two quantities that must be balanced in the upper bound theorem to determine an estimate of the collapse load.

**Line 133/134 in the new version: I think the sentence „Beyond 100000 elements, the meshing process …" can be omitted.**

The sentence was omitted.

**Line 230 new version: I would add an s: follows**

Correction has been made

**My comment to line 3 in the original version: Contrary to your statement you didn`t follow my suggestion. A model does not develop over a basal detachment.**

In the original version: "Line 3. I would omit the expression "in a kilometric-scale model" at this position or shift it to the end of the sentence in order to avoid strange associations (a model developed over a basal detachment)"
We apologize for the misunderstanding, originally, we omitted the sentence "in a kilometric-scale model" but we did not actually understand that the reviewer wanted an adjustment to the second part of the sentence. We agree with the reviewer, a model does not develop over a detachment, the wordings have been adjusted.

**My comment to line 20/21 in the original version: The references are modified not in all cases (e.g. Lines 22, 26, 27, 28, 30, 31, 33, etc. in the NEW version)**

Again, we apologize for the error and we performed the needed corrections.

**My comment to line 236 in the original version: what I meant was just to speak of deviatoric stress instead of stress.**

Adjustments have been made.

**Captions Figs. 8 and 9: Still not correct, b and d refer to SD**

Corrections have been made

**My comment to line Line 515 in the original version: I don't see any changes. Comparing with the reference below it seems that it should read Krabbenhøft, K., Lyamin, A., instead of K. Krabbenhøft, A. L.**

Corrections have been made

**I just have a final remark considering the purpose and capability of models in general (I expect noaction on this one). I'm starting with three citations of yours: In line 266/267 of the new version you write: „We remind the reader that the values obtained may seem very high but they are merely the results of an optimization process through the use of realistic parameters. Nevertheless, these values remain possible in theory." Your comment to my comment to Line 364 in the original version: „In theory they are valid and so they should be found in reality (not considering different aspects that might alter their behavior)."Your comment to my comment on Fig. 11 in the original version: „It is normal for these values to be higher than the ones observed in reality." I think it is valid to draw conclusions from models within the framework the model is defined. And this is what you did and have inteded. But caution is needed if conclusions on nature are made based on models. The motivation for a model of course is nature. So I find it important to draw conclusions within the model set up and then discuss in what aspects the results and conclusions may or may not reflect nature and if not what may be reasons for that (wrong asumptions, omission of relevant processes, etc.). I like the following two phrases:**
**"Essentially, all models are wrong, but some models are useful." (George Box)**
**„The purpose of models is not to fit the data but to sharpen the questions." (Samuel Karlin)**

Even though the reviewer did not ask for any corrections to be made, we adjusted the discussion paragraph by adding new a new subsection in which we relate our observations to real field observations.

We hope that these corrections are up to the reviewers expectations!